# Active Learning with LLMs for Partially Observed and Cost-Aware Scenarios

**Nicolás Astorga, Tennison Liu, Nabeel Seedat & Mihaela van der Schaar**
DAMTP, University of Cambridge
Cambridge, UK
`nja46@cam.ac.uk`

## Abstract

Conducting experiments and collecting data for machine learning models is a complex and expensive endeavor, particularly when confronted with limited information. Typically, extensive *experiments* to obtain features and labels come with a significant acquisition cost, making it impractical to carry out all of them. Therefore, it becomes crucial to strategically determine what to acquire to maximize the predictive performance while minimizing costs. To perform this task, existing data acquisition methods assume the availability of an initial dataset that is both fully-observed and labeled, crucially overlooking the *partial observability* of features characteristic of many real-world scenarios. In response to this challenge, we present Partially Observable Cost-Aware Active-Learning (POCA), a new learning approach aimed at improving model generalization in data-scarce and data-costly scenarios through label and/or feature acquisition. Introducing $\mu$POCA as an instantiation, we maximize the uncertainty reduction in the predictive model when obtaining labels and features, considering associated costs. $\mu$POCA enhance traditional Active Learning metrics based solely on the observed features by generating the unobserved features through Generative Surrogate Models, particularly Large Language Models (LLMs). We empirically validate $\mu$POCA across diverse tabular datasets, varying data availability, acquisition costs, and LLMs.

## 1 Introduction

In real-world machine learning (ML) applications, *fully-observed*, pristine training data is an exception rather than the norm. This challenge is especially evident during the initial stages of model development when training data is limited and varies in its informativeness across samples [1–3]. At this stage, obtaining additional data is crucial for improving model generalization but is fraught with challenges [4–6]. In particular, acquiring new data can be costly, often resulting in only essential features and labels being collected, leading to *partially observed* features in training data. Therefore, it's vital that acquisition is efficient, yet it remains unclear which features and labels from each instance will ultimately prove essential. Furthermore, data sources themselves can also be *partially observed*, with different features available across samples, further complicating the acquisition process. These challenges emphasize the importance of a new problem we call *Partially Observable Cost-Aware Active-Learning (POCA)* illustrated in Figure 1. Before its formalization in Section 2, we provide an intuitive overview:

> *"In situations with limited labeled data and partial feature observations, our objective is to enhance the generalization capabilities of a predictive model by strategically collecting features and/or labels. This goal should account the cost associated with data collection, as well as the varying levels of informativeness of labels and features across different instances"*

38th Conference on Neural Information Processing Systems (NeurIPS 2024).

Addressing the POCA problem is vital when building systems with partial observation or relevant features are yet to be defined, particularly in fields like customer churn, monitoring, healthcare, and finance (see Appendix A). For example, developing a churn customer prediction system might start with some basic client information, such as demographics and income. However, to build such a system, additional features may be needed, which could be gathered through further customer interactions or surveys. At the outset, it's uncertain which specific features will prove essential, and acquiring additional information and relevant labels (e.g., churn events) necessary to refine the ML system involves costs related to money, time, or risks limiting the data acquisition in practice. From a practical perspective, we envision POCA to be useful in applications or fields where missing features exist, and also data acquisition techniques like Active Learning (AL) are necessary. Applications from different fields dealing with missing features and/or applying AL can be found in Table 2.

**Related work.** The most related data acquisition technique is AL [7–10]. **AL** centers around enhancing model generalization through the acquisition of *only* additional labels. It operates under the assumption of having access to an initial small, fully observed training set (referred to as the historical labeled set) and seeks to acquire additional labels for samples from an unlabeled dataset (referred to as the pool set). This set is assumed to be *fully observed* in features, missing only labels. The distinctions between POCA and AL are illustrated in Figure 1. Tangentially, Active Feature Acquisition (AFA) methods [11–13] have been proposed to enhance the prediction of individual samples at test time—where the sample is partially observed. Like AL, it assumes a fully-observed historical labeled set, on

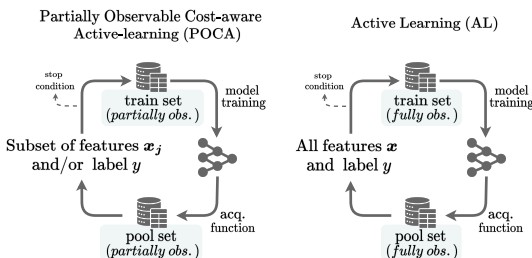

Figure 1: **Overview of data acquisition methods**. POCA acquires features and/or labels from a partially-observed pool incorporating them into a partially-observed training set. In contrast, AL targets label acquisition assuming a fully-observed pool set and training set.

which a model has already been trained. Given the trained model, the task then becomes identifying the most relevant unobserved features to acquire for partially observed instances at test time. We emphasize that AFA's primary focus is on optimizing feature acquisition for individual test samples, differing from our broader goal of data collection to enhance model training.

**Towards an Instantiation of POCA.** Given this problem definition, it is natural to wonder whether traditional AL metrics can be employed straightforwardly in the POCA setting. These metrics are usually derived from a predictive model that typically operates with fixed-size inputs. Consequently, predictions on partially observed instances can adversely affect the accuracy of AL metric estimations, leading to acquiring poor quality samples [14]. To overcome this challenge, we incorporate Generative Surrogate Models (GSM) to impute missing features in partially observed inputs, facilitating a more precise estimation of AL metrics. The effectiveness of GSM hinges on its ability to discern feature interrelations from available but unlabeled data. This task is particularly challenging due to the varying degrees of missingness in the instances and the constraints of limited sample sizes. To address these complexities, we employ Large Language Models (LLMs) to instantiate GSMs, utilizing their generation ability based on arbitrary conditioning and strong sample efficiency, allowing robust imputations to support the estimation of AL metrics under partial observability [15–17]

*Uncertainty* **POCA.** We term this instantiation *uncertainty* POCA ($\mu$POCA), due to its connection with Bayesian Experimental Design [18–22], and its application in Bayesian Active Learning (BAL)[7, 23] and Bayesian Optimization (BO)[24–26]. From a Bayesian perspective, $\mu$POCA maximizes the expected information gain or also known as expected uncertainty reduction, in the model's hypothesis resulting from an experiment. More specifically, $\mu$POCA extends the concepts of expected information gain in the model's parameters (EIG) and expected predictive information gain (EPIG) to partially observed scenarios, introducing PO-EIG and PO-EPIG, respectively [7, 27, 28]. Here, these methodologies maximize the expected uncertainty reduction when acquiring labels and a subset of features. Since the impact of unacquired features cannot be directly assessed, GSMs facilitate the computation of these metrics.

In summary, we make the following contributions:

① We address the unexplored challenge of costly data acquisition to enhance model generalization in partially observed scenarios. This leads us to introduce and formalize POCA, a novel ML paradigm for the acquisition of features and/or labels in the partially observed setting.

② We propose $\mu$POCA, a cost-aware Bayesian instantiation of POCA that maximizes the uncertainty reduction when acquiring data. $\mu$POCA extends traditional AL metrics by imputing partially observed instances using GSMs. We theoretically show that the uncertainty reduction is larger than using vanilla AL metrics.

③ We propose the use of LLMs as a specific instance of GSMs, designed to address challenges in partially observed scenarios, including data efficiency, arbitrary information conditioning, and handling both categorical and numerical feature values.

④ We empirically demonstrate $\mu$POCA outperforms standard active learning on a variety of partially observability scenarios spanning datasets, sample availability, and acquisition metrics—highlighting the usefulness and applicability of $\mu$POCA.

## 2  POCA: Partially Observable Cost-Aware Active-Learning

**Preliminaries.** Partially Observable Cost-Aware Active-Learning is a data acquisition problem that focuses on improving the predictive performance of $p_\phi(y|\boldsymbol{x})$ in the supervised setting, with $\phi$ the models we can employ. We denote $\boldsymbol{x} \in \mathcal{X}$ and $y \in \mathcal{Y}$ as instances of observed features and target, alongside the respective random variables (RV) $\boldsymbol{X}$ and $Y$. Bold variables, expressed as $\boldsymbol{x} = \{x_j\}_{j=1}^J$, represent a set of variables, in this case, features indexed by $j \in [J] = \{1, \ldots, J\}$, where the **bold form** of $j$ indicates a set of sub-indices $\boldsymbol{j}$. The sample index $i \in [I] = \{1, \ldots, I\}$, representing possible indexes in the pool set, is omitted when unnecessary, i.e., $x_{i,j} \equiv x_j$. We denote $\boldsymbol{x_o}$ as the observed features with $\boldsymbol{o} \subseteq [J]$.[1] In the general case, we assume that each *feature* $x_{i,j}$ considered for acquisition and the output of interest $y_i$ have associated acquisition costs $c_{i,j}$ and $c_{i,J+1}$. Here, $c_{i,\boldsymbol{j}}$ represents the total cost of acquiring the variables indexed by $\boldsymbol{j}$ for instance $i$.

> **POCA**
>
> In the context of *partially observed* data, our focus is on efficiently gathering features and/or labels to optimize a utility function, $U_t(\cdot)$ subject to an acquisition constraint $r_t(\cdot)$ at iteration $t$. $U_t(\cdot)$ quantifies the trade-off between the costs of data acquisition and the increased generalization capabilities of the model $\phi$, estimated from the available information $\boldsymbol{x_o}$ and the hypothetical acquisition of a specific set of features and/or labels. We formulate the optimization of this utility as follows:
>
> $$(i, \boldsymbol{j})^* = \underset{i \in [I], \boldsymbol{j} \subseteq [J+1]}{\arg\max} U_t(i, \boldsymbol{j}), \text{ s.t. } r_t(i, \boldsymbol{j}). \qquad (1)$$

$U_t(\cdot)$ is broadly defined, potentially estimated as result of using Bayesian techniques [18, 23, 27], frequentist techniques [29–31], RL techniques [32, 33], or can even be subjectively defined through human desires. Note, optimizing $U_t(\cdot)$ involves an iterative process of ① selecting the instance and variables $(i, \boldsymbol{j})^*$ to acquire (features and/or labels); ② adding these variables into the training set; ③ updating the model $\phi$ using the updated training set. In a more general case, this could also encompass batch acquisition [34, 35] by using $\boldsymbol{i}$ instead of $i$. Note that Eq. (1) represents the most general form of POCA, supporting model generalization when only features are acquired, as in semi-supervised or self-supervised learning. Our specific $\mu$POCA instantiation (Section 2.1) focuses on the supervised case, where selected features **and** labels are acquired.

**Common modalities for POCA.**    We anticipate that most applications of POCA will center on the tabular domain (see Table 2). However, it could also find valuable uses in fields like medical and satellite imaging, where noise-induced occlusion is common. In these cases, determining when a sample requires additional information (features) is essential for enhancing prediction accuracy and model training. Likewise, interactive robots that learn through vision may benefit from this approach, as they need to discern which scenarios (samples) merit interaction to effectively learn the relationship between features (objects) and labels (task to solve).

---

[1] In contrast with AL, POCA assumes $\boldsymbol{x_o} \subseteq \boldsymbol{x}$ instead of the fully observed assumption of AL $\boldsymbol{x_o} \equiv \boldsymbol{x}$. In addition, POCA considers the acquisition of features and/or labels, in this case, represented as $\boldsymbol{j}$.

## 2.1 $\mu$POCA: A Bayesian implementation of POCA

Although several techniques can be used to implement POCA, we opt for a Bayesian approach due to its widespread success in data acquisition literature. Building on the foundational principles of *Bayesian Experimental Design* [18, 23, 27], which provides a comprehensive framework for integrating various sources of information [36, 37], we introduce an instantiation of POCA within a Bayesian framework. This new approach, termed *uncertainty* POCA or $\mu$POCA, leverages information theory [38] to recast Eq. (1) as a cost-aware uncertainty reduction problem [27, 39]. The core of $\mu$POCA is centered on reducing uncertainty through a class of models that are exclusively trained using supervised learning, focusing on feature **and** label acquisition in partially observed scenarios. Here, we denote $\hat{\mu}_{i,j}$ as the uncertainty reduction for acquiring the label and features $j$ for sample $i$, which varies based on the approximation or method used.

> **$\mu$POCA**
>
> We reformulate the optimization problem (1) by substituting $U_t$ with a utility function $\tilde{U}$. This function $\tilde{U}$ is designed to capture the trade-off between uncertainty reduction, $\hat{\mu}_{i,j}$, and the acquisition costs associated with features and labels, represented by $\bar{c}_{i,j}$. The new objective can be expressed as:
> $$(i, j)^* = \underset{i \in [I], j \subseteq [J]}{\arg\max} \, \tilde{U}(\hat{\mu}_{i,j}, \bar{c}_{i,j}), \quad \text{s.t. } r(i, j), \tag{2}$$

here $\bar{c}_{i,j} = c_{i,j} + c_{i,J+1}$. In our research, we explore one specific instantiation, among potentially infinite options, denoted by $\tilde{U}_{i,j} = \hat{\mu}_{i,j}$ and $r(i, j) = \bar{c}_{i,j} < c$, respectively, with $c$ indicating the iteration's budget.[2]

**How to obtain this uncertainty?** We aim to minimize *epistemic uncertainty* [40, 41] by acquiring data, decreasing the predictive uncertainty produced by the possible hypothesis explaining the data. We work within the supervised model framework, hence we represent hypotheses as distributions over parameters. Our approach assumes the predictive model $p_\phi(y|x')$ can be expressed as:

$$p_\phi(y|x') = \mathbb{E}_{p_\phi(\omega)}[p_\phi(y|x', \omega)], \tag{3}$$

where $\omega \in \mathcal{W}$ is an instance of the parameter space and $\Omega$ its associated RV. Here, $\phi$ specifies the model choice, defining the functional form of $p_\phi(\omega) = p(\omega|\mathcal{D})$, the posterior given the observed training set $\mathcal{D}$, and the posterior predictive distribution $p_\phi(y|x')$, marginalized over $\omega$. Here, $x'$ represents a partially observed input, so estimating $p_\phi(y|x')$ must be adaptable to varying lengths of $x'$. To achieve this flexibility, models capable of handling variable-length inputs (such as Transformers) or, more broadly, marginalization techniques introduced in Section 3.2 can be employed.

This formulation is general, encompassing Bayesian models, neural networks with certain stochastic parameters [42, 43], and ensemble models [44, 45]. It also applies to Gaussian processes [46] when the posterior $p(\omega|\mathcal{D})$ is interpreted as a distribution over functions.

## 3 Method: Optimizing $\mu$POCA

The challenge in optimizing $\mu$POCA is in developing an uncertainty reduction metric, $\hat{\mu}_{i,j}$, that accurately represents the decrease in uncertainty when acquiring a subset of features $j$ for instance $i$, which has not been thoroughly investigated in the Bayesian literature. To address this, let's first provide some key background information. For data acquisition in ML, the primary focus has been on maximizing the expected uncertainty reduction, also known as expected information gain, when acquiring data [27]. This concept can be mathematically defined as:

$$\text{I}(A, B) := \text{H}(A) - \text{H}(A|B), \tag{4}$$

where $\text{H}(A)$ quantifies the uncertainty (entropy) about $A$, and $\text{H}(A|B)$ represents the uncertainty of $A$ after observing $B$ (in expectation). Existing AL approaches that utilize the expected reduction of uncertainty are summarized in Table 1. These methods maximize the uncertainty reduction of $\text{I}(\mathcal{G}, Y|\bullet)$ when $Y$ is observed.

---

[2]Alternative utility functions may balance uncertainty against costs as $\tilde{U}_{i,j} = \hat{\mu}_{i,j}/\bar{c}_{i,j}$. Other constraints could consider $c$ as the overall experimental budget.

Here, we use $\mathcal{G}$ to represent any random variable aligned with the generalization capabilities of the model and ● any arbitrary conditioning. In the Appendix, for completeness, we derive the estimation for these acquisition metrics.

Table 1: AL metrics with form of $\mathrm{I}(\mathcal{G}, Y | \bullet)$.

| Method | $\mathcal{G}$ | ● | objective |
|---|---|---|---|
| BALD [7] | $\Omega$ | $\boldsymbol{x}_o, \mathcal{D}$ | min. parameter uncertainty |
| EPIG [28] | $Y_{eval}$ | $\boldsymbol{x}_o, \mathcal{D}, \boldsymbol{X}_{eval}$ | min. predictive uncertainty |
| JEPIG [47] | $Y_{eval}^i$ | $\boldsymbol{x}_o, \mathcal{D}, \boldsymbol{X}_{eval}^i$ | min. predictive uncertainty |

### 3.1 Metrics for uncertainty reduction in *partially observed* scenarios

**Challenges in designing $\hat{\mu}_{i,j}$.** In real-world scenarios, the challenge is estimating uncertainty reduction based solely on accessible data $\boldsymbol{x}_o$. Traditional AL acquisition metrics, denoted as $\mu_\phi(\boldsymbol{x}_o)$, estimate uncertainty scores assuming $\boldsymbol{x}_o \equiv \boldsymbol{x}$. However, in partially observed scenarios where only a subset of inputs, $\boldsymbol{x}_o \subseteq \boldsymbol{x}$, is available, the observed features may lack sufficient informativeness for precise $y$ estimates and reliable uncertainty scores $\mu_\phi(\boldsymbol{x}_o)$.

**Generative Surrogate Model (GSM) to estimate metrics.** A more accurate estimate of current metrics can be achieved using the aforementioned AL metrics by imputing the potential missing features in expectation:

$$\mu_{\phi,\theta}^{\boldsymbol{j}}(\boldsymbol{x}_o) := \mathbb{E}_{\tilde{\boldsymbol{x}}_{\boldsymbol{j}}}[\mu_\phi(\boldsymbol{x}_o \cup \tilde{\boldsymbol{x}}_{\boldsymbol{j}})]. \qquad (5)$$

Here, the samples $\tilde{\boldsymbol{x}}_{\boldsymbol{j}}$ are obtained with a GSM denoted as $p_\theta(\boldsymbol{x}_{\boldsymbol{j}}|\boldsymbol{x}_o)$, which sample possible unobserved features $\boldsymbol{x}_{\boldsymbol{j}}$ based on the observed $\boldsymbol{x}_o$. It's worth noting that training $p_\theta(\boldsymbol{x}_{\boldsymbol{j}}|\boldsymbol{x}_o)$ could be done leveraging unlabeled data. In Figure 2, we illustrate the acquisition process of $\mu$POCA using GSMs.

**Why generative imputation can help Active Learning?** In Bayesian active learning, acquisition is closely linked to the concept of uncertainty reduction. To identify which features need to be acquired, it is essential to estimate the possible un-

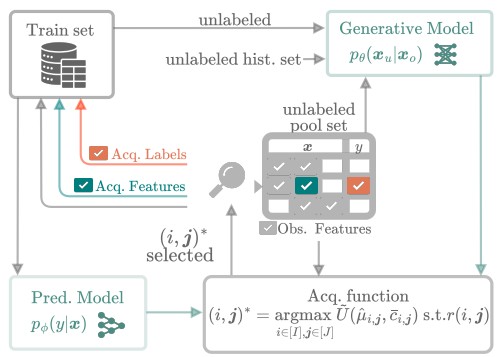

Figure 2: $\mu$POCA leverages GSMs trained on unlabeled data for imputing missing features. The imputed observations are used as an input for the predictive model, whose outputs are used to compute the acquisition metric.

observed values. If these values lie in areas of high uncertainty within the hypothesis space, acquiring these features is beneficial, as it will help reduce this uncertainty. Conversely, if the possible values for certain unobserved features show little or no impact on uncertainty, then acquiring these features may not be necessary. Notably, deterministic imputation cannot achieve this, as the lack of variability prevents assessment of its effect on uncertainty within the hypothesis space. This concept is illustrated in Figure 11 from Appendix H.

**Are we doing better?** We demonstrate the theoretical value of this approach for a family of acquisition metrics presented in Table 1, delving into their impact on the optimization process. These propositions convey the intuitive idea that acquiring more information, in this case, features, leads to a higher reduction in uncertainty for the predictive model (proofs can be found in Appendix B).

**Proposition 1.** *Let $\mu(\boldsymbol{x}_o)$ be an acquisition metric that can be written as $\mathrm{I}(\mathcal{G}, Y | \bullet)$, with $\mathcal{G}$ and ● representing the same variables observed in traditional AL (Table 1), and with $Y$, $\boldsymbol{X}_{\boldsymbol{j}}$ as previously defined. If $\mathcal{G} \perp\!\!\!\perp \boldsymbol{X}_{\boldsymbol{j}} | \bullet$, the following equality holds:*

$$\mathrm{I}(\mathcal{G}, (Y, \boldsymbol{X}_{\boldsymbol{j}}) | \bullet) = \mathbb{E}_{\boldsymbol{x}_{\boldsymbol{j}}} \mathrm{I}(\mathcal{G}, Y | \boldsymbol{x}_{\boldsymbol{j}}, \bullet) \qquad (6)$$

**Corollary 1.** *Under the assumptions of Proposition 1, the subsequent inequality is established:*

$$\mathbb{E}_{\boldsymbol{x}_{\boldsymbol{j}}} \mathrm{I}(\mathcal{G}, Y | \boldsymbol{x}_{\boldsymbol{j}}, \bullet) \geq \mathrm{I}(\mathcal{G}, Y | \bullet) \qquad (7)$$

*Equality is attained when $\mathcal{G} \perp\!\!\!\perp \boldsymbol{X}_{\boldsymbol{j}} | Y, \bullet$.*

🔍 Proposition 1 states that the *uncertainty reduction* of $\mathcal{G}$ (e.g., the random variable of the parameters, $\Omega$) by knowing $Y$ and $\boldsymbol{X}_{\boldsymbol{j}}$ is equivalent to the expected *uncertainty reduction* achieved by knowing $Y$ while conditioning on unobserved variables $\boldsymbol{x}_{\boldsymbol{j}}$. This is convenient as the conditioning on $\boldsymbol{x}_{\boldsymbol{j}}$ can be computed using Monte-Carlo approximation [48].

🔍 Corollary 1 implies that acquiring both *labels* and *features* results in greater uncertainty reduction compared to acquiring only *labels*, the objective maximized in traditional AL (Table 1). The uncertainty reduction is equivalent when, given ● and $Y$, the unobserved features $X_j$ don't have any impact in generalization $\mathcal{G}$.

Note that the independence assumption of Proposition 1 is valid in the supervised models we consider. In essence, this is because acquiring features without labels do not aid parameter updates and in consequence generalization improvements. The foundation of this assumption lies in the predictive mapping process from $X \rightarrow Y \leftarrow \Omega$, rather than in the data itself. Appendix B provides a more detailed explanation of this independence assumption's validity. Additionally, empirical evidence supporting the validity of Corollary 1 and, by extension, Proposition 1, is shown in Appendix K.

Equations (6) and (7) always apply to the true random variable of unobserved features or any of its approximations. However, the terms in Eq. (6) reflect the uncertainty reduction of obtaining the actual features when the approximated distribution of the GSM accurately reflects the distribution of the true random variable. We empirically investigate this approximation and its practical utility.

**PO Active learning metrics.** Building on Proposition 1 and Corollary 1, we extend BALD and EPIG as ▶ *Partially Observable Expected Information Gain (PO-EIG)*: $\mathbb{E}_{x_j} \mathrm{I}(\Omega, Y | x_j, x_o, \mathcal{D})$ and ▶ *Partially Observable Expected Predictive Information Gain (PO-EPIG)*: $\mathbb{E}_{x_j} \mathrm{I}(Y^{eval}, Y | x_j, x_o, X^{eval}, \mathcal{D})$. Corollary 1 states that these metrics provide a higher uncertainty reduction than their vanilla counterparts. We use Monte-Carlo for estimation (see Appendix C).

### 3.2 Predictive models in the PO setting

Our derivations are based on a distribution perspective, considering different numbers of conditioned variables. For instance, when calculating PO-EIG, expressed as $\mathbb{E}_{x_j} \mathrm{I}(\Omega, Y | x_j, x_o, \mathcal{D})$, it is necessary to compute the distribution $p_\phi(y | x_o, x_j)$. Here, $x_o$ could vary in length from one instance to another and $x_j$ varies based on the number of features considered for computing the uncertainty reduction metric. In practical terms, this means that the predictive model, attempting to approximate this distribution, must effectively handle inputs with varying variables and lengths.

To address this challenge, we employ GSMs to impute the missing information to enable predictive models that expect fixed-size inputs. This imputation is separated in two different steps (1) *conditioning* and (2) *marginalization*. Essentially, when evaluating the uncertainty reduction of an unobserved subset of features $x_j$ considered for acquisition, we *condition* on this subset $x_j$ and $x_o$ (the observed features), *marginalizing* over the remaining subset of unobserved features $x_{j'}$ (where $x_j \cup x_{j'}$ is the set of all unobserved features). This approximation process is mathematically formalized as follows, with supplementary visual aids provided in Figure 8 of Appendix C.3:

$$p_\phi(y|x_o, x_j) = \int p_\phi(y|x_o, x_j, x_{j'}) p_\theta(x_{j'}|x_o, x_j) = \mathbb{E}_{p_\theta(x_{j'}|x_o, x_j)}[p_\phi(y|x)], \qquad (8)$$

Here the predictive model simulates the behavior, wherein the predictive model only has access to $x_o$ and $x_j$ but it is computed using a model *as it would have all the features*. The marginalization step is essential for accurate metric estimation in the pool set and can also be applied during training. However, to reduce costs, we use GSM to impute features not acquired in the training set.

### 3.3 Efficient computation of utility function, Eq. (2)

Our goal is to maximize $\tilde{U}_{i,j}(\cdot) \equiv \tilde{U}(\hat{\mu}_{i,j}, \bar{c}_{i,j})$, which incorporates the uncertainty reduction $\hat{\mu}_{i,j}$. It is crucial to recognize that $\hat{\mu}_{i,j}$ could encompass all possible combinations of unobserved features. However, computing $\hat{\mu}_{i,j}$ for every possible combination of $(i, j)$ is impractical, since it is of order $\mathcal{O}(2^J)$. To overcome this challenge, we propose estimating the uncertainty reduction for all unobserved features and subsequently excluding the less relevant ones, i.e. those contributing minimally to uncertainty reduction. This ensures that we always retain the most relevant

---

**Algorithm 1** Acquisition process

1: $P = [\,], F = [\,]$
2: **for** $i \in [I]$ **do**
3: $\quad j^* = [J]$
4: $\quad$ **while** $r(i, j^*)$ **do**
5: $\quad\quad v^* = \arg\max_{v \in j^*} \tilde{U}_{i,j^* \backslash v} \quad$ s.t. $r(i, j^* \backslash v)$
6: $\quad\quad j^* = j^* \backslash v^*$
7: $\quad$ **end while**
8: $\quad P.\mathrm{add}(\hat{\mu}_{i,j^*}), F.\mathrm{add}(j^*)$
9: **end for**
10: $i^* = \arg\max_{i \in [I]} P[i], j^* = F[i^*]$
11: **Return:** $(i^*, j^*)$

features until the constraint $r$ in Eq. (2) is satisfied, in order $\mathcal{O}(J^2)$. The acquisition process is summarized in Algorithm 1, with feature selection steps highlighted in teal. Appendix C.3 provides details on an efficient approach to computing the *marginalization* step necessary for estimating $\hat{\mu}_{i,j}$. This efficiency can be further improved by selecting the most informative samples, followed by the application of Algorithm 1 (see Appendix D). For a comprehensive overview, including cost analyses, and details on GSM training and sampling, refer to Appendix D.

### 3.4 Large Language Models as Generative Surrogate Models

**LLMs as GSMs.** For the scenarios outlined in POCA, we specify the following desiderata for GSMs: (P1) generative capability, (P2) ability to learn from partially observed data, (P3) sample efficiency, and (P4) seamless integration of mixed-type variables. We argue that LLMs are well-suited to meet these criteria due to their ▶ generative capabilities and flexibility in training under ▶ arbitrary conditioning contexts [49–51]. Moreover, recent research highlights their exceptional performance in ▶ few-shot settings [52, 53] and their generative capabilities applied to ▶ tabular data comprising mixed-type attributes [51]. These strengths provide strong justification for focusing our research on LLMs as GSMs. However, **any** imputation method that fulfills these criteria may also serve as a suitable GSM, as further discussed in Appendix G.

We use LLMs as GSMs leveraging the unlabelled information via **Supervised Fine-Tuning (SFT)**. When working with tabular data, we serialize rows of the data, thereby converting it to natural language. For example, a set of features is serialized as *"Age is 25, Gender is Female, . . . , Blood pressure is 0.57"*. The LLM is then used to predict unobserved features based on available information. To achieve this, we utilize SFT on the LLM with the available observed features. The training data can encompass all unlabeled data, including historical and pool set data. The process entails generating random masks to form an input, $m \odot \boldsymbol{x}_o$, and an output, $(1 - m) \odot \boldsymbol{x}_o$, for SFT across all available data. This empowers the LLM to predict missing information by leveraging various combinations of observed features.[3] For more details, refer to Appendix F.2.

**Analysis of GSMs.** The effectiveness of $\mu$POCA in partially observed settings is closely tied to the GSM's ability to approximate the distribution of unobserved features. Two primary factors influence the accuracy of this approximation: **(1) the approximation capacity of the GSM** and **(2) intrinsic characteristics of the dataset**. A detailed examination of these factors is provided in Appendix J.

## 4 Experiments

We evaluate $\mu$POCA across three dimensions [4]: First, in the case that all features are acquired, we demonstrate that $\mu$POCA acquisition metrics are more informative in selecting instances with informative features than AL metrics. Second, we present a synthetic experiment accompanied by theoretical insights. Finally, we explore scenarios with budget constraints demonstrating that $\mu$POCA on more challenging scenarios.

Comparing $\mu$POCA with the current AL models is complex, as the latter are designed for fixed-size inputs. To address this challenge, we developed *Scenario 1* (see visual aid in Appendix H). This scenario involves dividing each instance in the pool set into the same observed and unobserved feature sets. We specifically select half of the features to remain unobserved, chosen by their high relevance to the predictive task as identified by a preliminary RF. It is important to note that while $\mu$POCA methods can handle any form of missing data, *Scenario 1* ensures a fair comparison by allowing AL models to operate without any modification, which could bias our evaluation. This scenario presumes the availability of a historical unlabeled dataset for training the GSM, using instances that include data on unobserved features. In practical applications, the pool set can often serve as the training set itself, representing a more realistic scenario we may encounter. We refer to this setting as *Scenario 2*. Results for this scenario are presented in Appendix I, where GSM is trained on partially observed data, while vanilla AL employs deterministic imputation to manage this case.

We select Magic, Adult, Housing, Cardio, and Banking tabular datasets based on their use in AL [28], tabular generation [49, 51], LLM-based classification [54], and relation with potential real-world

---

[3]Without loss of generality, in-context learning is viable for an LLM-based GSM

[4]Code can be found at: `https://github.com/jumpynitro/POCA` or `https://github.com/vanderschaarlab/POCA`

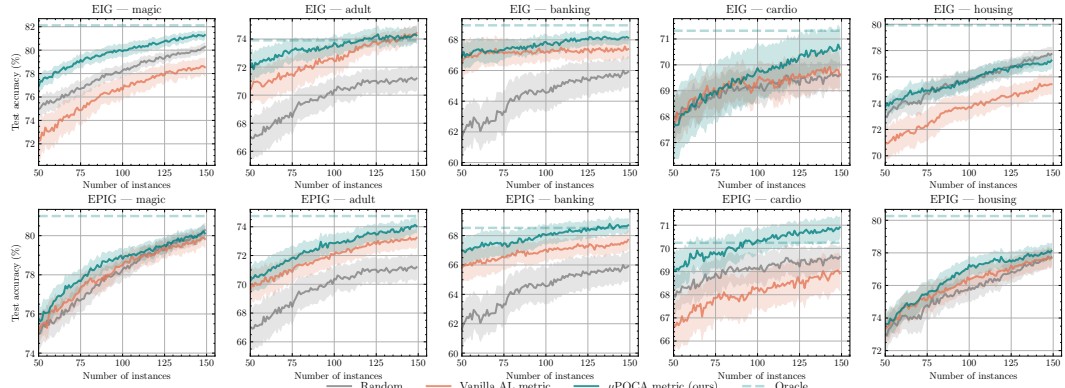

Figure 3: PO-EIG and PO-EPIG computed across diverse datasets - showing they either outperform or match their fully observed counterpart in terms of predictive performance

applications (Appendix A). These datasets have diverse characteristics: sample size, number of features, number of categorical, and numerical variables. We prioritize datasets with over 1000 samples to guarantee sufficient samples for the pool set. We showcase results using a RF trained with 100 estimators. We start training with two fully observed samples per class, conduct 150 acquisition cycles, repeat each experiment over 60 seeds, and display a 95% confidence interval. We train Mistral7B-Instruct-v0.3 using 8 Monte-Carlo samples for generative imputation.

## 4.1 Need for POCA: Shortfalls of Active Learning

**Objective.** To assess the need for more generalized methodologies such as $\mu$POCA, we analyze the performance of PO-EIG, a partially observed extension of BALD (EIG)—the most widely used metric in active learning literature. Additionally, we incorporate EPIG into the study, a recently developed active learning metric within the 1 family. According to our theoretical framework (see Corollary 1), *PO-metrics* outperforms their vanilla counterparts in terms of uncertainty reduction. Our goal is to examine whether this uncertainty reduction leads to improved downstream performance when all features are acquired based on the same information, $x_o$, or, in other words, if the selected instances possess features that are more relevant.

**Setup.** To ensure a fair comparison, we evaluated *PO-metrics* and *Vanilla-metrics* under *Scenario 1*, using Random and Oracle as reference baselines. Here, *Oracle* represents the *Vanilla-metrics* acquisition metric, but with access to all features. Ideally, when GSM functions optimally, the performance of PO-EIG should align with that of *Oracle*.

**Analysis**. The first thing to note is that EIG metrics computed with partially observed features can be significantly worse than simple baselines like random as shown in Magic dataset from Figure 3) (top). Figure 3 (top) demonstrates that PO-EIG generally either *outperforms* or worst case matches their fully observable counterparts BALD across all datasets. A similar behavior is observed for PO-EPIG, which generally outperforms their vanilla metric counterpart. This suggests that an increase in uncertainty reduction translates into an increase in downstream performance. While PO-EIG and PO-EPIG metrics consistently outperform baselines, they occasionally fall short of oracle performance, notably in the Housing datasets. This may stem from two factors: Firstly, the GSM has poor prediction performance on the unobserved data due to insufficient data or model capacity. Secondly, even with adequate capacity and data, weak correlation between unobserved data and the target hinders the acquisition process. We study these factors in Appendix J. We note that it is non-trivial to quantify the GSM's capability or correlations of unobserved data to the target. Thus, the practical implication is that both *PO-metrics* should be preferred in PO settings, providing a performance boost or at least matching their *vanilla* counterparts. Additionally, in Appendix I.1, we include other relevant Active Learning metrics that, while not fitting into the family of studied metrics, also demonstrate performance gains with the proposed framework.

> 💡 First, AL metrics computed on partially observed features can dramatically fail for selecting relevant instances. Second, PO-EIG and PO-EPIG generally *outperform* or match fully observed counterparts.

## 4.2 Theoretical insights

**Objective.** We investigate the implications of our theoretical findings (Eq. (7)) on the acquisition process; determining whether a weak correlation between unobserved features and the target, results in a small gap between PO-EIG and BALD. We also explore how correlation affects performance.

**Setup:** We create an intuitive synthetic 2D experimental setup (Figure 4) with a variable target. The target is determined linearly with varying slopes, leading to different correlations with the features. Our chosen features—$X_1$, $X_2$, and $X_3$—represent data along the x-axis, y-axis, and a Gaussian category, respectively. Introducing the Gaussian category injects stochasticity into the marginalization process, ensuring non-trivial solutions. The observed feature is $X_1$, with possible acquisition of $X_2$, $X_3$. We examine three scenarios: 1) Low Corr($X_2$,$Y$), where the class depends solely on $X_1$ due to vertical slope; 2) High Corr($X_2$,$Y$), where the class depends solely on $X_2$, rendering $X_1$ irrelevant; and 3) Mid. Corr($X_2$,$Y$), where $X_1$ has some impact. Note, we evaluate acquisition metrics and performance until the convergence of the oracle (BALD with all features)

**Analysis.** Figure 5A empirically validates that PO-EIG is always equal to or greater than BALD, consistent with our theoretical insights (Eq. (7)). Figure 5B illustrates the evolution of the metric gap between PO-EIG and BALD under varying correlations between $X_{2,3}$ and $Y$. In low correlation scenarios (orange line), the gap diminishes towards the acquisition's end, aligning with Corollary 1 where both metrics should converge when $\mathcal{G} \perp\!\!\!\perp \boldsymbol{X}_{2,3}|x_{1,\bullet}$, i.e., when the unobserved features don't impact generalization. Initially, the gap exists as the model learns from data the redundancy of unobserved features. The same figure shows larger correlation leads to a wider gap, observed most notably in the large correlation scenario (purple) and moderately in the medium correlation scenario (teal). Figure 5 shows that, generally, the degree of problem correlation provides a proxy correlation with acquisition performance. For example, the purple line exhibits the largest difference between BALD and EIG. Particularly in low correlation scenarios, the performance difference between PO-EIG and BALD is negligible across the acquisition (orange line).

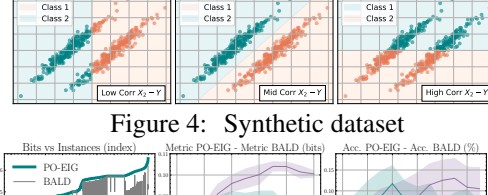

Figure 4: Synthetic dataset

Figure 5: Comparing PO-EIG and BALD.

## 4.3 Cost-aware active learning

**Objective:** We evaluate the performance of $\mu$POCA (specifically PO-EIG) under budget-constrained feature acquisition, aiming to determine if acquiring only a subset of features, denoted as $\boldsymbol{j}$, offers an advantageous trade-off in performance. This selective acquisition approach enables acquiring a larger number of instances within the same budget. Furthermore, we aim to show that imputation alone cannot fully replace the need for direct data acquisition.

**Setup.** We use the Magic dataset as a case study to examine the impact of cost constraints on predictive performance and the feature acquisition process. To facilitate this assessment, we introduce costs associated with both features and labels. For simplicity and visualization clarity, we assume the cost of an instance to be 1, representing the sum of the costs for all features and the label, with each feature assigned an equal cost. This setup allows us to analyze four distinct approaches: (1) the Vanilla acquisition metric (EIG), (2) PO-EIG, (3) PO-EIG with a maximum feature acquisition limit of 60%, and (4) PO-EIG with unrestricted feature acquisition. We evaluate the performance of these approaches in three ways: by accuracy based on acquired instances (Figure 6, left), by performance relative to the budget utilized (assuming no label costs) (Figure 6, middle), and by performance with varying label costs under a fixed total budget of 50 (Figure 6, right).

**Analysis.** Figure 6 (left) illustrates that acquiring fewer features generally results in decreased performance; however, it still outperforms the EIG baseline. While limited feature acquisition impacts performance, it allows for a more efficient budget allocation across instances, enabling the acquisition of a larger instance pool. This trend is visible in Figure 6 (middle), where performance is plotted against the total budget spent, assuming no label cost. Here, methods focusing on selective feature acquisition excel, as they gather more overall information through increased instance count and key features.

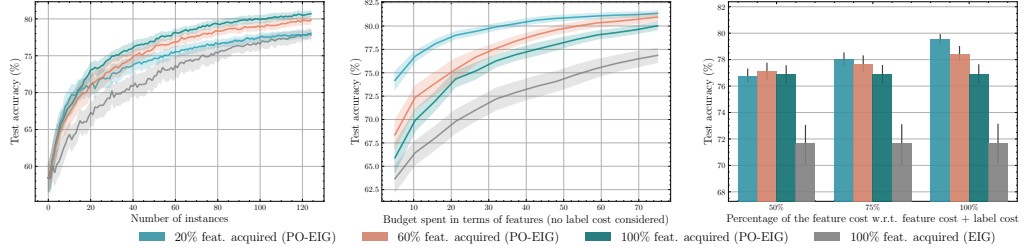

Figure 6: **Left:** Accuracy vs. number of instances acquired. **Middle:** Accuracy vs. budget without considering label costs. **Right:** Accuracy vs. budget with varying label costs.

Figure 6 (right) demonstrates that the optimal PO-EIG method depends on feature acquisition cost: when cost is heavily weighted toward feature acquisition (right histogram), the best method is PO-EIG with 20% of features acquired, whereas a 50% label cost favors PO-EIG with 60% feature acquisition. While these findings might suggest that acquiring fewer features and imputing the rest is optimal for maximizing instances, this approach may introduce noise into the training set, potentially biasing the model. To explore this, we analyze model performance at different levels of feature acquisition in Figure 7, with varying levels of pool data, using the full pool set for training (excluding non-acquired features). As shown on the y-axis, acquiring more features enhances performance. When the budget is unlimited, acquiring all available data is preferable; however, in practice, this may not be feasible, making POCA approaches advantageous.

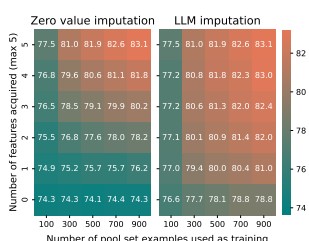

Figure 7: Acquiring vs imputing.

> ♀ First, $\mu$POCA metrics (PO-EIG) can be more cost-effective than common active learning metrics. Second, imputation is useful for missing data but shouldn't replace data acquisition.

## 5 Discussion

We introduce and formalize POCA a data acquisition framework, addressing the vital but underexplored challenge of partially observed settings. Through $\mu$POCA, a practical implementation of this framework, we demonstrate the feasibility of acquiring unobserved features and labels based on those partially observed features, using more generalized AL utility metrics — computed by estimating features generated using an LLM-based GSM. Our results over various scenarios are substantially more effective than alternatives — of substantial value for data acquisition in cost-restrictive environments. We hope the POCA framework and our subsequent findings will spur additional work to advance data acquisition in partially observed settings.

**Limitations.** Our work focuses on the values of features, providing a general framework where restrictions are the main source of constraints in terms of acquisition. However, we do not assess how these restrictions are selected, which could be a promising area for future research. We also note that we use LLMs in the context of data acquisition. Like any GSM, LLMs can indeed exhibit biases that affect the acquisition process. In this study, we did not consider this issue, and it represents an interesting avenue for future work. If necessary, current debiasing techniques can be applied.

**Practical consideration and future work.** (1) In the PO setting with data "missingness," GSM imputation is essential for acquisition. Future work could quantify uncertainty [43, 55] to assess GSM efficacy. (2) LLM capability also impacts acquisition; while we use a 7B-parameter model, larger models could further enhance performance, though this is beyond our current scope.

## Acknowledgments and Disclosure of Funding

We thank the anonymous NeurIPS reviewers, members of the van der Schaar lab, and Andrew Rashbass for insightful comments and suggestions. NA thanks W.D. Armstrong Trust for sponsorship and support. TL thanks AstraZeneca for support. NS thanks the Cystic Fibrosis Trust. This work was supported by Microsoft's Accelerate Foundation Models Academic Research initiative.

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

# Appendix: POCA: Partially Observable Cost-Aware Active-Learning with Large Language Models.

## A   Appendix A: Real-World Use Cases

Table 2: **Real-world use-cases of POCA.** We outline real-world scenarios where the POCA framework can have an impact. For each problem domain, we describe partially observable features, labels, and the underlying predictive task. We categorize references into three types: A) where active learning is employed, B) where predictive modelling is performed in the presence of partially observed features, and C) active learning is applied to partially observed settings (with data pre-processing to handle missing features). The symbol ▶ stands for **acquisition costs**.

| Problem Setting | Observed Features | Acquirable Features | Possible Labels / ML Task | References |
| --- | --- | --- | --- | --- |
| Customer Churn | Basic customer data (demographics, plan type, usage patterns). | Detailed customer interaction data and satisfaction surveys ▶ data collection and operational costs. | Churn events. ▶ Risk, analysis of customer status over time. | A: [56, 57], B: [58] |
| Marketing and Consumer Research | Consumer demographics, basic purchase history. | Consumer preferences via surveys, social media activity ▶ survey deployment and data processing. | Purchase decisions or brand perception changes. ▶ market analysis or consumer feedback mechanisms. | A: [59, 60], B: [61, 62] |
| Finance | Basic financial information (income level, employment status, existing debts), market trends. | Credit history, detailed investment portfolios. ▶ operational costs, data acquisition from external agencies and privacy concerns. | Loan defaulting, investment outcomes ▶ Risk (time required for outcomes to manifest and the analysis needed.) | A: [63–66], B: [67–69] |
| Healthcare Diagnostics (Medicine) | Basic patient information (demographics, medical history, basic vitals). | Results from specific medical tests (blood tests, MRI scans, etc.). ▶ Medical test costs/operation costs. | Diagnosis of specific diseases ▶ Clinical evaluation, Expert analysis or medical tests that could be more expensive than acquiring features. | A:[70–76], B:[77–80], C: [70, 81] |
| Predictive Maintenance in Manufacturing | Regular operation data (machine runtime, temperature, vibration levels). | Detailed inspections or advanced sensor data (acoustic emissions, ultrasonic testing). ▶ operational costs. | Failure events or maintenance needs ▶ Risk for not doing mantainence. Inspection or equipment failure costs. | A: [82–86], B:[87–89] |
| Customized E-commerce Recommendations | User activity (page views, clicks), basic demographics | Detailed purchase history, and product review text. Also consumer preferences via surveys, social media activity ▶ Survey deployment | Recommendation ▶ Risk of wrong recommendation | A: [90–92], B:[93, 94] |
| Environmental Monitoring | Basic weather data (temperature, humidity, precipitation), satellite imagery. | Results from specific sensor data (soil moisture, specific pollutant levels) ▶ Operational costs of measuring data | Environmental condition classifications ▶ field surveys, lab analysis of samples | A: [95], B [96] |

Table 2 illustrates that active learning is extensively utilized in a variety of real-world application scenarios. Furthermore, it is not uncommon in these contexts to encounter situations with incomplete data, which can harm generalization [2, 6, 97, 98] capabilities of downstream models. The breadth of related work covers diverse sectors including customer churn prediction, marketing research, healthcare diagnostics, and predictive maintenance. While active learning is adept at selectively querying labels in scenarios where data is fully observed, its application in the context of missing data is less clear. The challenge is compounded by the fact that, in similar problem settings, it is not always guaranteed that features will be fully observed. This reality underscores the need for alternative machine learning techniques to address such challenges. Our proposed approach, POCA, offers a novel solution for applying active learning in scenarios with partially observed data, taking into account realistic cost constraints.

# B  Appendix B: Acquisition metrics for partially observed scenarios.

**Proposition 1.** *Let $\mu(\boldsymbol{x}_o)$ be an acquisition metric that can be written as $\mathrm{I}(\mathcal{G}, Y|\bullet)$, with $\mathcal{G}$ and $\bullet$ representing the same variables observed in traditional AL (Table 1), and with $Y$, $\boldsymbol{X_j}$ as previously defined. If $\mathcal{G} \perp\!\!\!\perp \boldsymbol{X_j}|\bullet$, the following equality holds:*

$$\mathrm{I}(\mathcal{G}, (Y, \boldsymbol{X_j})|\bullet) = \mathbb{E}_{\boldsymbol{x_j}} \mathrm{I}(\mathcal{G}, Y|\boldsymbol{x_j}, \bullet) \tag{9}$$

*Proof:* We can decompose the left part of Eq. (9) as:

$$\mathrm{I}(\mathcal{G}, (Y, \boldsymbol{X_j})|\bullet) = \mathrm{I}(\mathcal{G}, \boldsymbol{X_j}|\bullet) + \mathrm{I}(\mathcal{G}, Y|\boldsymbol{X_j}, \bullet) \tag{10}$$

Using $\mathcal{G} \perp\!\!\!\perp \boldsymbol{X_j}|\bullet \implies \mathrm{I}(\mathcal{G}, \boldsymbol{X_j}|\bullet) = 0$, the first term on the right of Eq. (10) cancels, obtaining $\mathrm{I}(\mathcal{G}, (Y, \boldsymbol{X_j})|\bullet) = \mathrm{I}(\mathcal{G}, Y|\boldsymbol{X_j}, \bullet) = \mathbb{E}_{\boldsymbol{x_j}} \mathrm{I}(\mathcal{G}, Y|\boldsymbol{x_j}, \bullet)$ concluding the proof.

**Corollary 1.** *Under the assumptions of Proposition 1, the subsequent equivalent inequality is established:*

$$\mathbb{E}_{\boldsymbol{x_j}} \mathrm{I}(\mathcal{G}, Y|\boldsymbol{x_j}, \bullet) \geq \mathrm{I}(\mathcal{G}, Y|\bullet) \tag{11}$$

*Equality is attained when $\mathcal{G} \perp\!\!\!\perp \boldsymbol{X_j}|Y, \bullet$.*

*Proof:* Symmetrically as before we can decompose the left part of Eq. (9) as:

$$\mathrm{I}(\mathcal{G}, (Y, \boldsymbol{X_j})|\bullet) = \mathrm{I}(\mathcal{G}, Y|\bullet) + \mathrm{I}(\mathcal{G}, \boldsymbol{X_j}|Y, \bullet) \tag{12}$$

Using proposition (1), we obtain:

$$\mathbb{E}_{\boldsymbol{x_j}} \mathrm{I}(\mathcal{G}, Y|\boldsymbol{x_j}, \bullet) = \mathrm{I}(\mathcal{G}, Y|\bullet) + \mathrm{I}(\mathcal{G}, \boldsymbol{X_j}|Y, \bullet) \tag{13}$$

Clearly the equality is obtained when $\mathcal{G} \perp\!\!\!\perp \boldsymbol{X_j}|Y, \bullet$ since the mutual information is zero. When taking $\mathrm{I}(\mathcal{G}, \boldsymbol{X_j}|Y, \bullet) \geq 0$ we obtain:

$$\mathbb{E}_{\boldsymbol{x_j}} \mathrm{I}(\mathcal{G}, Y|\boldsymbol{x_j}, \bullet) \geq \mathrm{I}(\mathcal{G}, Y|\bullet), \tag{14}$$

concluding the proof.

**Observation:** Note that, the independence assumption $\mathcal{G} \perp\!\!\!\perp \boldsymbol{X_j}|\bullet$ always hold for the class of supervised learning models we consider. $\mathcal{G}$ is a random variable representing the generalization capabilities of the model $\omega \in \mathcal{W}$, with random variable $\Omega$. This random variable is subject to model training $\mathcal{A}$, which can be written as the mapping between the training set $\mathcal{D}$ and hyperparameters $h \in \mathcal{H}$ to the output $\omega$, i.e., $\mathcal{A}: \mathcal{D} \times \mathcal{H} \to \mathcal{W}$. Additionally, the model prediction is a mapping $\mathcal{P}$ between the model $\omega$ and the input $x \in X$ to the output $Y$, i.e., $\mathcal{P}: \mathcal{W} \times \mathcal{X} \to \mathcal{Y}$.

It is crucial to acknowledge that the "world generator" influences $\mathcal{D}$, $X$, and $Y$, but does not directly affect $\Omega$. Given that $\mathcal{D}$ is observable, any connection through this path is cut. The sole connection of $\Omega$ to $X$ and $Y$ is through the mapping $\mathcal{P}$, which establishes a causal structure: $\Omega \to Y \leftarrow X$. According to this structure, $\Omega$ and $X$ are generally independent unless $Y$ is observed, leading to a dependence due to $\mathcal{P}$ creating a configuration known as an "immorality" among these variables. This explains why $\mathcal{G} \perp\!\!\!\perp \boldsymbol{X_j}|\bullet$ holds; however, this independence may not persist in scenarios where $\mathcal{G} \perp\!\!\!\perp \boldsymbol{X_j}|\bullet, Y$.

In the context of our work the input $X$ can be decomposed in $X = x_o, X_j$ the observed part of the random variable and the unobserved part of the random variable. Consequently, $X_j$ follows the same independency assumptions of $X$.

## C Appendix C: Monte-Carlo Estimates

In our study, we evaluate BALD and EPIG, along with their partially observed counterparts, PO-EIG and PO-EPIG. For completness, we show the estimatation the vanilla metrics, and later their corresponding partially observed extensions.

### C.1 PO-EIG and BALD

We follow a similar notation to [28]. For categorical variables, BALD can be decomposed as:

$$\mathrm{I}(\Omega, Y | \boldsymbol{x_o}, \mathcal{D}) = \mathrm{H}(y | \boldsymbol{x_o}, \mathcal{D}) - \mathrm{H}(y | \boldsymbol{x_o}, \Omega, \mathcal{D}) \tag{15}$$

$$= \mathbb{E}_{p_\phi(\omega)} \left[ \mathbb{E}_{p_\phi(y|x)}[\log p_\phi(y|x)] + \mathbb{E}_{p_\phi(y|x,\omega)}[\log p_\phi(y|x,\omega)] \right] \tag{16}$$

$$\approx - \sum_{y \in \mathcal{Y}} \hat{p}_\phi(y|\boldsymbol{x_o}) \log \hat{p}_\phi(y|\boldsymbol{x_o}) + \frac{1}{K} \sum_{k=1}^{K} \sum_{y \in \mathcal{Y}} p_\phi(y|\boldsymbol{x_o}, \omega_k) \log p_\phi(y|\boldsymbol{x_o}, \omega_k), \tag{17}$$

where $K$ represent the total number of parameter samples $\omega_k \sim p_\phi(\omega)$ from the posterior distribution given $\mathcal{D}$. Here,

$$\hat{p}_\phi(y|\boldsymbol{x_o}) = \frac{1}{K} \sum_{k=1}^{K} p_\phi(y|\boldsymbol{x_o}, \omega_k). \tag{18}$$

PO-EIG extends this formulation to include missing features as conditioning samples $\tilde{\boldsymbol{x}}_{\boldsymbol{j}}$ from the GSM, accommodating partially observed settings. To illustrate, let's first consider the case where the metric is conditioned over all unobserved features, i.e., $\boldsymbol{j'} = \emptyset$, which correspond to results of Figure 3. Following a similar decomposition, we can approximate $\mathbb{E}_{\tilde{\boldsymbol{x}}_{\boldsymbol{j}}} \mathrm{I}(\Omega, Y | \boldsymbol{x_o}, \mathcal{D}, \tilde{\boldsymbol{x}}_{\boldsymbol{j}})$ as:

$$\mathbb{E}_{\tilde{\boldsymbol{x}}_{\boldsymbol{j}}} \mathrm{I}(\Omega, Y | \boldsymbol{x_o}, \mathcal{D}, \tilde{\boldsymbol{x}}_{\boldsymbol{j}}) \approx \sum_{l=1}^{L} \left[ - \sum_{y \in \mathcal{Y}} \hat{p}_\phi(y|\boldsymbol{x}^l) \log \hat{p}_\phi(y|\boldsymbol{x}^l) \right.$$
$$\left. + \frac{1}{K} \sum_{k=1}^{K} \sum_{y \in \mathcal{Y}} p_\phi(y|\boldsymbol{x}^l, \omega_k) \log p_\phi(y|\boldsymbol{x}^l, \omega_k) \right], \tag{19}$$

here $L$ represent the total number of Monte-Carlo samples from the GSM, with $\tilde{\boldsymbol{x}}_{\boldsymbol{j}}^l$ one possible sample. Here,

$$p_\phi(y|\boldsymbol{x}^l) = p_\phi(y| \underbrace{\boldsymbol{x_o}, \tilde{\boldsymbol{x}}_{\boldsymbol{j}}^l}_{\boldsymbol{x}}) \tag{20}$$

$$\hat{p}_\phi(y|\boldsymbol{x}^l) = \frac{1}{K} \sum_{k=1}^{K} p_\phi(y|\boldsymbol{x}^l, \omega_k). \tag{21}$$

**What happen when $j$ doesn't consider all unobserved features?**. This scenario is useful when we want to asses the impact acquiring of subset of the unobserved features. Utilizing a smart notation, this approximation can be stated identically as Equation (19), but the estimation of the predictive distribution changes slightly:

$$p_\phi(y|\boldsymbol{x}^l, w_k) = \sum_{p=1}^{P} p_\phi(y| \underbrace{\boldsymbol{x_o}, \boldsymbol{x}_{\boldsymbol{j}}^l, \boldsymbol{x}_{\boldsymbol{j'}}^p}_{\boldsymbol{x}}, w_k), \tag{22}$$

here, $P$ are a total of new MC samples from the GSM. This *marginalization* trick is necessary to deal with model that expect fixed size inputs.

## C.2 PO-EPIG and EPIG

For PO-EPIG and EPIG, we replace the sub-index $eval$ by $*$. Similar to before, for categorical variables, EPIG can be decomposed as:

$$I(Y_*, Y | \boldsymbol{x_o}, X_*, \mathcal{D}) = \mathbb{E}_{x_*}[I(Y_*, Y | \boldsymbol{x_o}, x_*, \mathcal{D})] \tag{23}$$

$$= \mathbb{E}_{x_*}[\text{KL}(p_\phi(y_*, y | \boldsymbol{x_o}, x_*) || p_\phi(y_* | x_*) p_\phi(y | \boldsymbol{x_o})] \tag{24}$$

$$\approx \frac{1}{M} \sum_{m=1}^{M} \sum_{y \in \mathcal{Y}} \sum_{y_* \in \mathcal{Y}_*} \hat{p}_\phi(y, y_* | \boldsymbol{x_o}, x_*^m) \log \frac{\hat{p}_\phi(y, y_* | \boldsymbol{x_o}, x_*^m)}{\hat{p}_\phi(y | \boldsymbol{x_o}) \hat{p}_\phi(y_* | x_*^m)}, \tag{25}$$

here,

$$\hat{p}_\phi(y, y_* | \boldsymbol{x_o}, x_*^m) = \frac{1}{K} \sum_{k=1}^{K} p_\phi(y | \boldsymbol{x_o}, \omega_k) p_\phi(y_* | x_*^m, \omega_k), \tag{26}$$

$$\hat{p}_\phi(y | \boldsymbol{x_o}) = \sum_{k=1}^{K} p_\phi(y | \boldsymbol{x_o}, \omega_k), \tag{27}$$

$$\hat{p}_\phi(y_* | x_*^m) = \sum_{k=1}^{K} p_\phi(y_* | x_*^m, \omega_k) \tag{28}$$

Similarly as before, PO-EPIG extends this formulation to include missing features when conditioning on samples $\tilde{\boldsymbol{x}}_{\boldsymbol{j}}$ from the GSM. We now derive the estimation of this metric when estimating PO-EPIG when considering all the unobserved features $\boldsymbol{j}$. We can decompose $\mathbb{E}_{\tilde{\boldsymbol{x}}_{\boldsymbol{j}}} I(Y_*, Y | \boldsymbol{x_o}, X_*, \mathcal{D}, \tilde{\boldsymbol{x}}_{\boldsymbol{j}})$ as:

$$\mathbb{E}_{\tilde{\boldsymbol{x}}_{\boldsymbol{j}}} I(Y_*, Y | \boldsymbol{x_o}, X_*, \mathcal{D}, \tilde{\boldsymbol{x}}_{\boldsymbol{j}}) \approx \sum_{l=1}^{L} \left[ \frac{1}{M} \sum_{m=1}^{M} \sum_{y \in \mathcal{Y}} \sum_{y_* \in \mathcal{Y}_*} \hat{p}_\phi(y, y_* | \boldsymbol{x}^l, x_{**}^m) \right.$$
$$\left. \times \log \frac{\hat{p}_\phi(y, y_* | \boldsymbol{x}^l, x_{**}^m)}{\hat{p}_\phi(y | \boldsymbol{x}^l) \hat{p}_\phi(y_* | x_{**}^m)} \right], \tag{29}$$

$$\hat{p}_\phi(y, y_* | \boldsymbol{x}^l, x_*^m) = \frac{1}{K} \sum_{k=1}^{K} p_\phi(y | \underbrace{\boldsymbol{x_o}, \tilde{\boldsymbol{x}}_{\boldsymbol{j}}^l}_{\boldsymbol{x}}, \omega_k) p_\phi(y_* | x_*^m, \omega_k), \tag{30}$$

$$\hat{p}_\phi(y | \boldsymbol{x}^l) = \sum_{k=1}^{K} p_\phi(y | \underbrace{\boldsymbol{x_o}, \tilde{\boldsymbol{x}}_{\boldsymbol{j}}^l}_{\boldsymbol{x}}, \omega_k), \tag{31}$$

$$\hat{p}_\phi(y_* | x_*^m) = \sum_{k=1}^{K} \sum_{h=1}^{H} p_\phi(y_* | x_*^m, \boldsymbol{x}_{\boldsymbol{j}}^h, \omega_k), \tag{32}$$

Note that the marginalization step of the "evaluation set" is also necessary since we also assume it to be partially observed. Here, $\boldsymbol{x}_{\boldsymbol{j}}^h$ are Monte-Carlo samples necessary make the predictive model receive fixed inputs. While expensive, PO-EPIG can also compute metrics when only a subset of features is estimated to be acquired. The trick is similar to PO-EIG, being necessary a marginalization across the $\boldsymbol{j}'$ for the the predictive mode receive fixed-size inputs (as (22)).

## C.3 Illustration of efficient MC estimation

As outlined in the earlier sections, we approximate the PO-metrics through the techniques of conditioning and marginalization. Figure 8 provides a visual representation of this approximation.

In Eq. (8), for efficient computation in the marginalization of $j'$, we substitute $p_\theta(\boldsymbol{x}_{j'}|\boldsymbol{x}_o, \boldsymbol{x}_j)$ with $p_\theta(\boldsymbol{x}_{j'}|\boldsymbol{x}_o)$, using the same samples used to compute the uncertainty reduction across all unobserved features. This approximation does not significantly compromise precision, as we exclude the least relevant feature sequentially. Consequently, the samples used to marginalize an "irrelevant" feature remain minimally impacted by the overall sampling strategy.

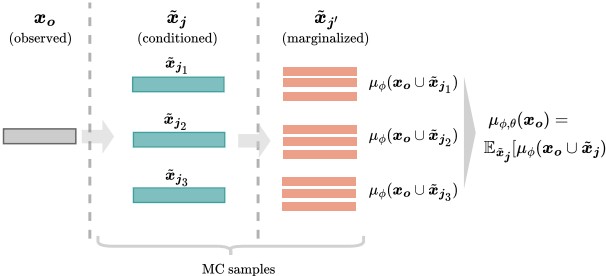

Figure 8: Illustrative diagram demonstrating the application of *conditioning* and *marginalization* techniques in the estimation of PO-metrics for an arbitrary instance.

# D  Pseudo-code of the whole acquisition process

---

**Algorithm 2** $\mu$POCA Algorithm

---

**Require:** Initial pool set $\mathcal{P}$ indexed initially by $[I]$, initial partially observed unlabeled dataset $\mathcal{D}_u$, initial partially observed labeled dataset $\mathcal{D}_l$, identify costs $c$ for every element in the pool set, identify number of features $J$, select downstream model $\phi$, select Generative Surrogate Model parameterized by $\theta$, select number of Monte-Carlo samples $S$. If using heuristic select the number of instances to analize for subset of feature selection.

1: $\theta \leftarrow \texttt{maximize\_likehood}(\theta, \mathcal{D}_u)$
2: $\mathcal{P}_S \leftarrow \texttt{generative\_imputation}(\theta, \mathcal{P}, S)$
3: **while** $\texttt{stop\_condition}(\cdot) == \texttt{False}$ **do**
4:      $\tilde{\mathcal{D}}_l \leftarrow \texttt{impute\_missing\_data}(\mathcal{D}_l, \theta)$       # Imputing when using a predictive model that receive
    fixed-size input
5:      $\tilde{\phi} \leftarrow \texttt{maximize\_likehood}(\phi, \tilde{\mathcal{D}}_l)$
6:      **if** $\texttt{use\_heuristic}$ **then**
7:          $I^* \leftarrow \texttt{compute\_top\_uncertain\_instances}(\tilde{\phi}, \mathcal{P}_S, [I], [J])$
8:      **end if**
9:      $P = [\,], F = [\,]$
10:     **for** $i \in [I]$ **do**
11:         **if** $\texttt{use\_heuristic}$ and $i \notin I^*$ **then**
12:             $P.\text{add}(-\inf), F.\text{add}([])$
13:             $\texttt{continue}$
14:         **end if**
15:         $\boldsymbol{j}^* = [J]$
16:         **while** $r(i, \boldsymbol{j}^*)$ **do**
17:             $U = [\,]$
18:             **for** $v \in \boldsymbol{j}^*$ **do**
19:                 **if** $r(i, \boldsymbol{j}^* \backslash v)$ **then**
20:                     $\hat{\mu}_{i,\boldsymbol{j}^*} \leftarrow \texttt{compute\_uncertainty}(\tilde{\phi}, \mathcal{P}_S, i, \boldsymbol{j}^* \backslash v)$
21:                     $U.\text{add}(\tilde{U}(\hat{\mu}_{i,\boldsymbol{j}}, \bar{c}_{i,\boldsymbol{j}}))$
22:                 **else**
23:                     $U.\text{add}(-\inf)$
24:                 **end if**
25:             **end for**
26:             $v^* = \arg\max U$
27:             $\boldsymbol{j}^* = \boldsymbol{j}^* \backslash v^*$
28:         **end while**
29:         $P.\text{add}(\hat{\mu}_{i,\boldsymbol{j}^*}), F.\text{add}(\boldsymbol{j}^*)$
30:     **end for**
31:     $i^* = \arg\max_{i \in [I]} P[i], \boldsymbol{j}^{**} = F[i^*]$
32:     $\mathcal{D}_l \leftarrow \mathcal{D}_l \cup \mathcal{P}[i^*][\boldsymbol{j}^{**}]$
33:     $\mathcal{P} \leftarrow \mathcal{P} \setminus \mathcal{P}[i^*]$
34:     $\mathcal{P}_S \leftarrow \mathcal{P}_S \setminus \mathcal{P}_S[i^*]$
35:     $I \leftarrow I - 1$
36: **end while**

---

We assume that GSMs will be trained using available unlabeled data, which may be either fully observed (if a bank of fully observed unlabeled data is available) or partially observed (using the pool set itself), respectively. This assumption enables us to train the GSM and generate imputed values for the pool set before the acquisition process begins. The complete acquisition is detailed in Algorithm 2.

We define the generation cost as $C_g$ and the downstream cost as $C_d$. Algorithm 2 indicates that the sampling cost for GSMs is $\mathcal{O}(I \cdot J \cdot S \cdot C_g)$, where $I$ is the number of instances, $J$ is the number of features, and $S$ represents the number of Monte Carlo samples. The inference cost of the downstream model is $\mathcal{O}(I \cdot J \cdot S \cdot C_d)$.

The cost of acquiring a subset of features depends on the restriction $r(i, \boldsymbol{j})$, which is bounded by $J$ (the case where $J$ features are discarded for acquisition). The cost for acquiring a subset of features for a single instance is $\mathcal{O}(J^2 \cdot S \cdot C_d)$ using Algorithm 1. To reduce this overhead, we first select the most informative instance (assuming all features are acquired) in $\mathcal{O}(I \cdot J \cdot S \cdot C_d)$ (L7 in Algorithm 2), and then select the subset of features to acquire using Algorithm 1 (10 in our case) in order $\mathcal{O}(J^2 \cdot R \cdot S \cdot C_d)$. This analysis shows that the most critical factor is the number of features $J$, as it affects both sampling and the downstream model (quadratically in this case).

Another consideration is when the available data is insufficient for a reliable GSM. In this case, any additional features acquired during the acquisition process can be used to update the GSM weights periodically, which increases the costs.

# E  Datasets

Datasets were constructed such the number of pool samples were numerous enough to determine the impact of the acquisition performance any confounding effect. The distribution of pool set were selected maintaining the distribution of the original dataset similar to previous work [28]. The "evaluation distribution" in EPIG is follows the same distribution of pool set. As mentioned in the main text, we maintain a small unlabeled historical set for GSM training allowing fair comparison with Active Learning. For large datasets like Adult and Housing we limit their maximum original data size avoiding unnecessary costs in Monte-Carlo sample generation. The test set is defined as the 30 % of the intial dataset.

- Magic [99]: Original data size of 19020 samples. Historical set of 1000 samples. Pool set distribution; Class0: 4980 samples. Class1: 2700

- Adult [100]. Original data size of 19020 samples (after cut). Historical set of 1000 samples. Pool set distribution; Class0: 5760 samples. Class1: 1920.

- Housing. Original data size of 19020 samples (after cut). Historical dataset of 1000 samples. Pool set distribution; Class0: 3840, Class1: 3840.

- Cardio [101]. Original data size of 100k samples. Historical dataset of 1000 samples. Pool set distribution; Class0: 3000, Class1: 3000.

- Banking [102] Original data size of 45211 samples. Historical dataset of 400 samples. Pool set distribution; Class0: 2000, Class1: 500.

We did slight modification in datasets allowing LLM better comprehension. For Magic, we didn't include fConc1 since its name similarity with fConc. For CMC dataset, we change categorical values 0 and 1 to the categories that were represented on metadata. For example, in column "Standard-of-living_index" instead of using 0,1, 2, 3 we use Low, Medium-Low, Medium-High, and High.

With this adjustment the final columns of these datasets are:

- Magic. 9 numerical columns: fLength, fWidth, fSize, fConc, fAsym, fM3Long', fM3Trans, fAlpha, fDist.

- Adult. 8 categorical columns: workclass, education, marital-status, occupation, relationship, race, sex, native-country. 6 numerical columns: age, fnlwgt, education-num, capital-gain, capital-loss, hours-per-week.

- Housing. 8 numerical columns: MedInc, HouseAge, AveRooms, AveBedrms, Population, AveOccup, Latitude, Longitude.

- Cardio. 10 numerical columns: ID, age, age_years, height, weight, ap_hi, ap_lo, bmi. 7 categorical variables: gender, cholesterol, gluc, smoke, alco, active, bp_category_encoded.

- Bank. 2 numerical columns: age, balance. 6 categorical variables: job, marital, education, contact, day_of_week, default.

For model training, we use one hot encoding for categorical variables.

# F  Large Language Models, training and evaluation

We analyzed primarily one open-source Mistral7B-Instruct-v0.3 with more in Section J. Preliminary findings show that instruct LLMs generally outperformed their non-instruct counterparts. In what follows, we give component used and a general overview of the prompts used:

- Input: The input is the tabular information provided to the LLM. Here, it is in the form of "{*feature A*: *value A*, *feature B*: *value B*, *feature C*: *value C*}", for possible features and values.

- Query: The query represents the information requested by the LLM to generate a response. It should only include the attribute names, formatted as follows: "{*feature D: [Mask]*, *feature E: [Mask]*, *feature F: [Mask]*".

- **Response**: The input is the tabular information provided to the LLM. Here, it is in the form of only values: "{*feature D*: *value D*, *feature E*: *value E*, *feature F*: *value F*}".

The specific prompts were adjusted to specific LLM format.

### F.1 Mistral7B-Instruct-v0.3

Training time: approximately 1-2 hours

{**Input, **Query} ###Response: {**Input, **Response}

Test time: The total number of generated samples varies depending on the dataset, with processing times ranging from 3 to 8 hours.

{**Input, **Query} ###Response:

Here as some samples and the respones On Magic:

{fAsym: -90.6431, fM3Trans: -10.169, fDist: 250.51, fConc: 0.1637, fAlpha: [Mask], fM3Long: [Mask], fSize: [Mask], fWidth: [Mask], fLength: [Mask]} ###Response: {fAsym: -90.6431, fM3Trans: -10.169, fDist: 250.51, fConc: 0.1637, fAlpha: 1.719, fM3Long: -33.7346, fSize: 3.3316, fWidth: 28.4232, fLength: 69.8763}

{fConc: 0.3174, fM3Trans: -4.5924, fAsym: -28.5264, fDist: 150.686, fSize: [Mask], fM3Long: [Mask], fWidth: [Mask], fAlpha: [Mask], fLength: [Mask]} ###Response: {fConc: 0.3174, fM3Trans: -4.5924, fAsym: -28.5264, fDist: 150.686, fSize: 2.6819, fM3Long: -21.4579, fWidth: 16.6296, fAlpha: 18.3037, fLength: 27.3688}

{fM3Trans: -19.0262, fDist: 294.682, fConc: 0.1546, fAsym: -137.729, fAlpha: [Mask], fLength: [Mask], fWidth: [Mask], fM3Long: [Mask], fSize: [Mask]} ###Response: {fM3Trans: -19.0262, fDist: 294.682, fConc: 0.1546, fAsym: -137.729, fAlpha: 0.294, fLength: 124.816, fWidth: 38.3973, fM3Long: -71.1274, fSize: 3.3655}

{fM3Trans: -11.0238, fDist: 85.2971, fConc: 0.1765, fAsym: 32.2464, fWidth: [Mask], fM3Long: [Mask], fAlpha: [Mask], fLength: [Mask], fSize: [Mask]} ###Response: {fM3Trans: -11.0238, fDist: 85.2971, fConc: 0.1765, fAsym: 32.2464, fWidth: 27.559, fM3Long: -51.3547, fAlpha: 21.297, fLength: 56.6845, fSize: 3.3454}

### F.2 Masking

We tested two masking strategies. First, when we had access to all the historical data, particularly when comparing against traditional AL in the main text. Second, when certain features might be missing, as encountered in Appendix I. In the first scenario, we randomly masked some observed information, always ensuring at least two features as input to prevent overly complex tasks for the LLM. In the second scenario, due to the partial observability of data, some features could be less observed than others, leading to varying degrees of missingness.

### F.3 Training specification

We train the models using QLoRA using 4 bit quantization, $r = 32$, lora_alpha= 64, for 10000 steps, with 2 samples per batch size, and a learning rate of 7.5e-5.

# G  Generative Surrogate Models

In this section, we outline the core desiderata for Generative Surrogate Models to be used within the $\mu$POCA framework and compares related imputation methods against the desiderata.

- **[P1] Generative capability**: The model must model a non-deterministic distribution over the unobserved features to effectively identify and prioritize the most relevant features. To exemplify, we consider $x_{j^*}$ as the subset of current features under review for acquisition, and $x_{j'}$ as the features already excluded from acquisition. PO-EIG's acquisition metric strategically selects the feature, $v^*$, that maximizes uncertainty reduction among all possible features, $v$, considered for exclusion from the set $j^*$. This effectively minimizes information loss. Formally, this is expressed as:

$$v^* = \arg\max_{v \in j^*} \mathbb{E}_{\tilde{x}_{j \setminus v}} I(\Omega, Y | \tilde{x}_{j \setminus v}, x_o, \mathcal{D}). \tag{33}$$

  In contrast, employing a deterministic GSM modifies the acquisition strategy for imputed features $\bar{x}$, simplifying to:

$$v^* = \arg\max_{v \in j^*} I(\Omega, Y | \bar{x}_{j \setminus v}, x_o, \mathcal{D}).$$
$$= \arg\max_{v \in j^*} I(\Omega, Y | \bar{x}, x_o, \mathcal{D}). \tag{34}$$

  The second equality arises from the fact that we are limited to a single value due to estimated conditioning and marginalization. Consequently, any permutation of conditioning sets $j$ and marginalization sets $j'$ will lead to the same metric for a deterministic GSM. (34) explicitly demonstrates that employing deterministic GSM results in an acquisition metric independent of $v$, the feature under consideration for exclusion through a greedy approach. In conclusion, the GSM should model a stochastic distribution for acquisition purposes.

- **[P2] Learning on partially observed data**: The GSM must learn from partially observed training data, where different instances will have varying observed features.

- **[P3] Sample-efficiency**: Given the potential scarcity and variability of feature observability of training data, the GSM must efficiently learn from limited samples with different observed features.

- **[P4] Supports mixed-type variables**: To support a broad range of data types, the GSM should enable generative modelling across both continuous and discrete variables.

Table 3: **Overview of imputation methods**. Comparison based on key desiderata of a GSM.

| Desiderata | LLM | Generative imputation | Discriminative imputation | Sample statistics |
|---|---|---|---|---|
| References | - | [103–109] | [110–116] | |
| **[P1]** | ✓ | ✓ | ✗ | ✗ |
| **[P2]** | ✓ | ✓ | ✓ | ✓ |
| **[P3]** | ✓ | ✗ | ✓ | ✗ |
| **[P4]** | ✓ | ✗ | ✗ | ✗ |

**Flexibility of Large Language Models.** We believe that Large Language Models hold significant potential as sample-efficient methods for feature estimation. Although LLMs are not specifically pretrained for tabular data estimation, their general capabilities have demonstrated effectiveness across various domains. These general abilities have been successfully applied in optimization [24, 117, 118], reasoning [119, 120], planning [121–123], concepts [124, 125], autonomous adaptation [126–128], digital twin construction [129, 130], meta-learning [131, 132], uncertainty reduction [133, 134], and as tabular data generators [49]. They have also shown promise as few-shot tabular learners in supervised settings [52, 53]. The application of LLMs in the intersection of tabular data tasks, particularly in the last two contexts, provides direct evidence of their viability in this area and to be used as GSMs in the context of data acquisition.

## H   Comparing with Active Learning

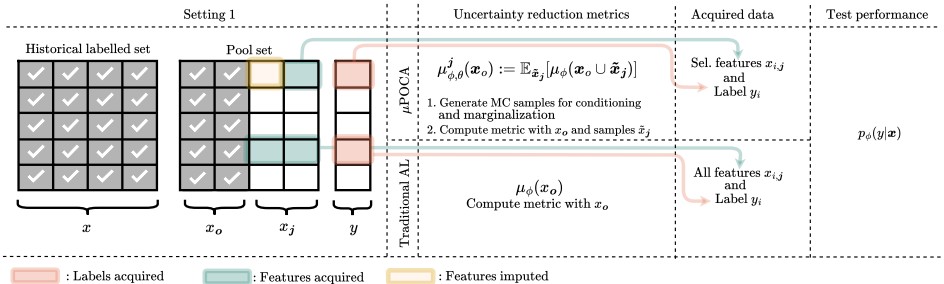

Figure 9: **Scenario 1.**

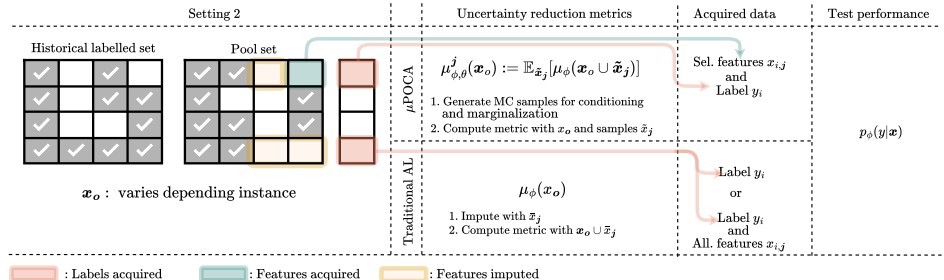

Figure 10: **Scenario 2.**

We used *Scenario 1* as indicated in the main text. Another scenario where traditional active learning metrics can be applied is *Scenario 2*, which utilizes conventional imputation methods. The results for this scenario are shown in Figure 12 when all features are acquired for the Active Learning metric. As discussed in the main paper deterministic imputation does not allowed for the acquisition of subset of features, which is illustrated in Figure 11.

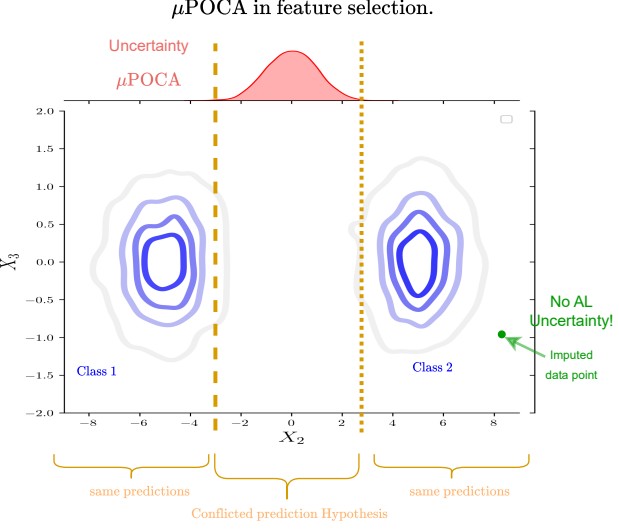

Figure 11: Figure (a) shows the distribution of $X_2$ and $X_3$ conditioned on $x_1$. With estimates of $X_2$ and $X_3$, $\mu$POCA can identify the relevant feature ($X_2$) and the relevant region. In contrast, AL metrics might use deterministic imputation (green), which does not reveal feature relevance or area of importance under partial observability. This is because a point estimate can not explore the $X_2$, $X_3$ and how their variability affects the outcome.

# I   Additional Results

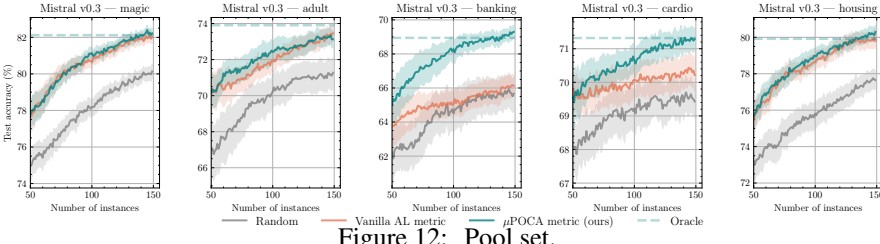

Figure 12:   Pool set.

Figure 12 illustrates that Mistral-Instruct-v0.3 trained LLMs solely on the Pool dataset, introducing a different scenario than the discussed in the main text. In this setup, only three features are consistently observed in the pool set, while others may be absent with uniform of probability. While the effectiveness of LLMs would depend on case-by case scenarios. In this demonstration, we underscore their viability and potential as Generative Surrogate Models (GSMs).

## I.1   Varying uncertainty metrics

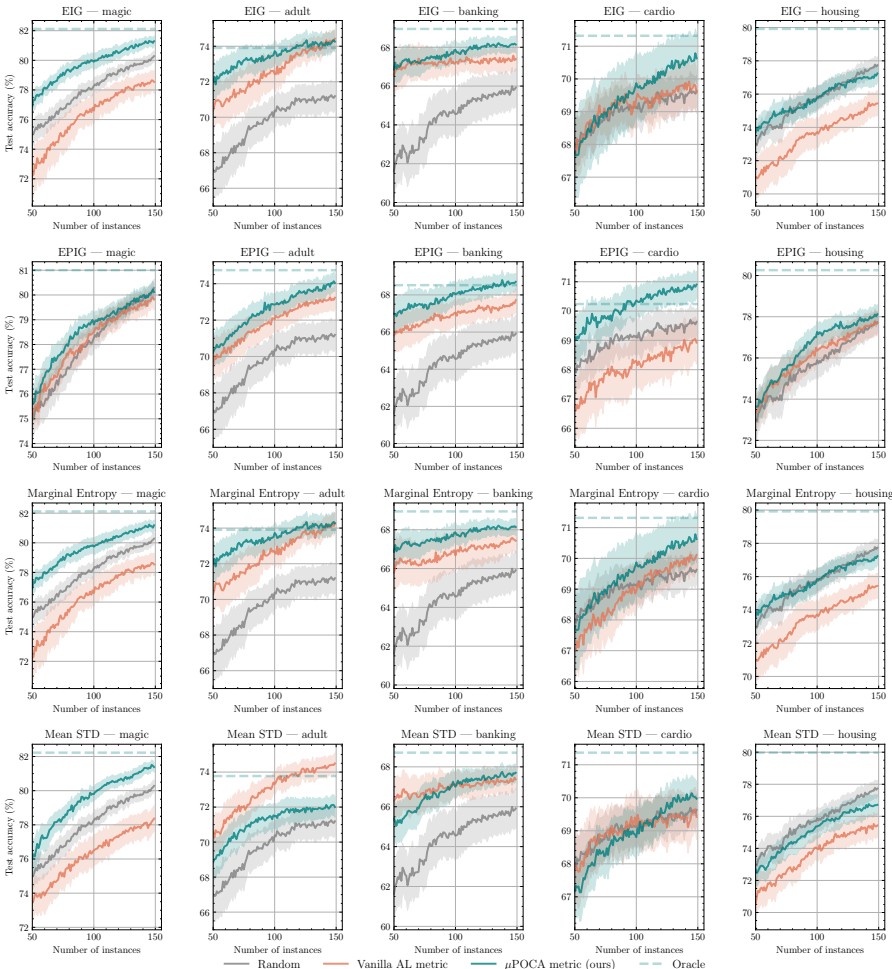

Figure 13:   Partially observed active learning metrics and their fully observed counterparts.

# J   Analysis of GSM

The performance of $\mu$POCA in partially observed settings fundamentally depends on how well the GSM approximates the distribution of the unobserved features. There are two key factors that affect the quality of this estimation:

- **GSM's approximation power**: Referring to the model's capacity to accurately model the unobserved features.

- **Intrinsic characteristics of the dataset**: Referring to inherent correlations between observed and unobserved features. Indeed, lower correlations are more challenging.

In what follows, we investigate each factor in turn, [G1] studying the impact of different GSMs and [G2] investigating different dataset characteristics. This approach aims to delineate the conditions under which $\mu$POCA is expected to excel.

## [G1] GSM impact on acquisition performance

**Setup.** We evaluated various LLMs (including Mistral-7B-Instruct-v0.3, Gemma2, and Llama-3.1) as GSMs to assess how model quality affects acquisition performance.

**Analysis.** Figure 14 demonstrates that GSM quality significantly influences acquisition results. Notably, Mistral-7B consistently outperforms the alternative GSMs, with one exception in the housing dataset. Interestingly, Gemma 2 performs well on this benchmark, highlighting this inter-model variability. The Cardio dataset further highlights these differences, with Mistral-7B performing significantly better than the other models.

**Takeaway.** These findings underscore the critical role of GSM quality in acquisition performance.

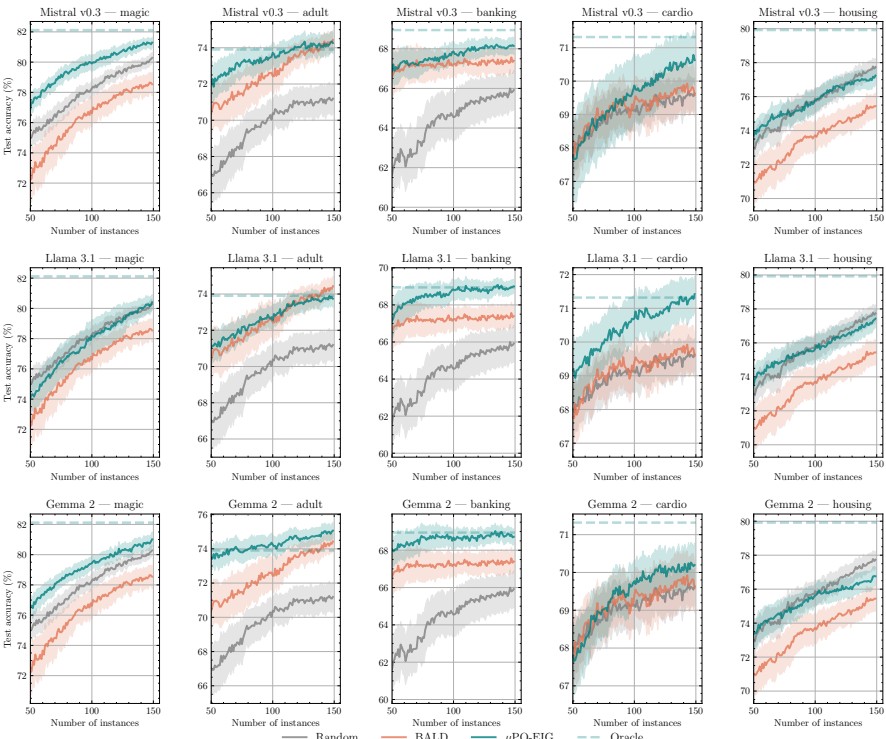

Figure 14:   Varying LLMs with Mistral7B-Instruct v0.3 based on EIG acquisition metric.

**[G2] Performance across varying data characteristics**

**Setup.** Next, we turn our attention to investigating how the data distribution affects acquisition performance. We are particularly interested in analyzing the effect of the correlation between unobserved features $X_{unobs}$ and observed features $X_{obs}$ on acquisition performance.

To demonstrate this, we examine a scenario where (1) $X_{unobs}$ correlates with the outcome, while (2) observed features do not. In this context, the GSM becomes crucial for downstream performance, as $X_{obs}$ alone provides insufficient information to predict outcomes accurately. As such, the GSM must effectively model the relationship between $X_{obs}$ and $X_{unobs}$ to acquire missing features critical for predicting the outcome.

We model both $X_{obs}$ and $X_{unobs}$ as two-dimensional random Gaussian variables centered at zero and establish a specific controllable correlation between them through $\rho$:

$$\Sigma = \begin{bmatrix} I_2 & \rho_{X_{obs}X_{unobs}} I_2 \\ \rho_{X_{obs}X_{unobs}} I_2 & I_2 \end{bmatrix}$$

The label $Y$ is then constructed to be independent of $X_{obs}$ using the orthogonalization:

$$X_{\text{orthogonal}} = X_{unobs} - X_{obs}(X_{obs}^T X_{obs})^{-1} X_{obs}^T X_{unobs}$$

which we use to construct the label using

$$\text{logits} = \frac{1}{1 + e^{-\sum X_{\text{orthogonal}}}}, \quad C = \mathbf{1}_{\text{logits}>0}$$

**Analysis.** Figure 15a illustrates how varying $\rho$ between $X_{obs}$ and $X_{unobs}$ empirically affects variable correlation, $c$, validating our synthetic experiment design. Figure 15b analyzes EIG (traditional active learning without GSM) and PO-EIG with varying $\rho$. We note that when the correlation between $X_{obs}$ and $X_{unobs}$ is low, GSMs provide no performance benefits. However, as correlation increases, the performance gains of PO-EIG over EIG expand significantly, confirming our hypothesis.

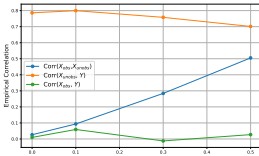

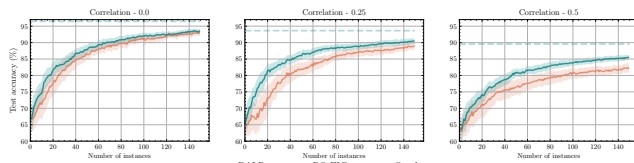

(a) Correlations between $X_{\text{obs}}$ and $Y$, and between $X_{\text{unobs}}$ and $Y$ is roughly unaffected across different vaues of $\rho$.

(b) Performance of EIG Across Datasets with Varying Correlations Between Observed Features ($X_{obs}$) and Unobserved Features ($X_{unobs}$). **Observation:** As correlation increases, the performance gains of PO-EIG (using GSM) over BALD (traditional AL) becomes more notable.

Figure 15: **Synthetic experiments.** Figure (a) visualizes the characteristics of the synthetic data with different values of $\rho$, while Figure (b) demonstrates the performance of EIG (BALD) and PO-EIG as the degree of correlation between observed and unobserved features varies.

# K    PO-EIG vs BALD

Figure 16 illustrates the comparison of uncertainty reduction between PO-EIG and BALD (EIG) at iteration 50 of training with seed zero. It is evident that PO-EIG consistently achieves equal or greater uncertainty reduction than EIG. This empirical observation supports the validity of Corollary 1, and consequently, substantiates the assumption made in Proposition 6.

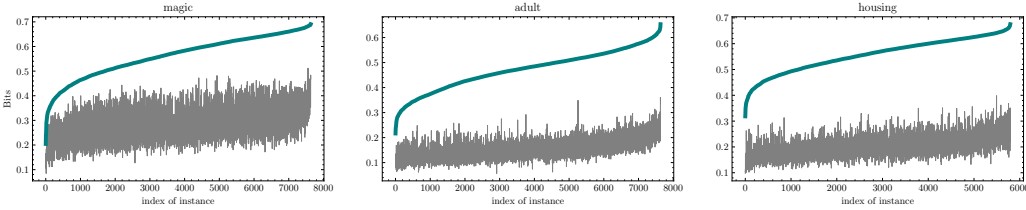

Figure 16:  PO-EIG vs BALD metrics on various scenarios at iteration 50.

