# OpenReview forum: "Active Learning with LLMs for Partially Observed and Cost-Aware Scenarios"
_NeurIPS.cc/2024/Conference — NeurIPS 2024 poster_

### Official Review · Reviewer_A8wS · 2024-07-08

**Soundness:** 4
**Presentation:** 4
**Contribution:** 3
**Rating:** 6
**Confidence:** 3

**Summary:**

This paper explores the problem of active learning where we can choose not just which data point to annotate, but also which features to obtain, when given an unlabeled pool of data points for which only some features are known. The paper proposes using generative models to sample the missing features and apply bayesian active learning techniques to estimate the expected information gain from annotating the data points.

**Strengths:**

The paper is overall strong, addresses an interesting problem and proposes a simple and intuitive methodology that extends standard techniques. The implications of the theory is discussed and validated with empirical analysis.

**Weaknesses:**

There are some points of clarification in writing and the numerical analysis is missing some potential interesting ablations and comparisons with different baselines. See questions below

**Questions:**

- What is the motivation for having the label cost be separate in problem (1)? If this problem is not really studied and if you are focusing only on (2) anyways where the label cost is baked in, then this seems unnecessary generalization.
- If you have only some features for some data, what is the training process that you use?
- How does the quality of the generative model affect the downstream performance of POCA? Have you tried different models, or at the minimum, injected artificial noise to estimates?
- The numerical analysis primarily focuses on comparisons against BALD. While this makes sense given that the method is an extension of BALD, it would be interesting to see how other benchmark active learning techniques fare on this problem. Specifically, could it be that other uncertainty or diversity-based active learning methods, beyond Bayesian ones, also perform well for example if the number of missing features is small?

**Limitations:**

Limitations are discussed in the paper

---

> ### Author Rebuttal · Authors · 2024-08-07
>
> *We are grateful to the reviewer for their insightful feedback and constructive comments that have improved the paper.*
>
> ---
> ## [P1] Inclusion of label cost in Equation 1
>
> Thank you for your concerns. We appreciate the opportunity to clarify the inclusion of label cost in Equation 1.
>
> In Equation 1, we include label costs because this expression allows for the acquisition of a subset of features without the label. This separation of label cost is intentional.
>
> Note that one of our primary goals is to formalize the POCA problem, and for this, we opted for the most generalized problem definition. Notably, the POCA problem, i.e., acquisition of a subset of features and/or labels, along with its associated cost, is not explicitly addressed in the existing experimental design literature **[C3]**, which is a contribution in itself as remarked by other reviewers (**CwS1**).
>
> The utility function in Equation 1 measures the level of generalization based on acquisition. Semi-supervised and self-supervised learning are examples of how generalization can be achieved using only unlabeled data.
>
> ---
> ## [P2] Training when some features are available
>
> Thank you for highlighting this concern. We can interpret this issue in two ways:
>
> - **Training a downstream model with only a subset of features.** In the context of $\mu$POCA, we work with predictive models that require fixed-size inputs. However, not all features may be observed due to the acquisition of a subset of features. To handle this situation, we employ a GSM to impute any missing features. This approach ensures that the downstream model receives the fixed-size inputs it requires. This imputation is indicated in L208.
>
> - **Training a GSM with only a subset of features.** This is a common case when the GSM is trained on the pool set (results in Figure 11). Note, that our training of the GSM for any available unlabeled data. As described in L246, the process involves generating random masks to create an input $m \odot \boldsymbol{x}_o$, and an output, $(1 - m) \odot \boldsymbol{x}_o$. However, we ensure the input always has at least two features and the output at least one feature. Therefore, we only consider training samples with at least three observed features. This detail is now included in Appendix E.2.
>
> As illustrated in the $\mu$POCA Algorithm (see attached PDF), the predictive model is trained after each data acquisition. In contrast, the GSM is trained with the available unlabeled data before the acquisition process begins.
>
> ---
> ## [P3] GSM quality
>
> We wish to clarify the impact of GSM quality on downstream performance. The ability of GSMs to accurately approximate the underlying data distribution is crucial for effective generative feature imputation, which in turn affects both predictive accuracy and acquisition performance in downstream tasks. In **[G1, G2]**, we identify the specific conditions under which GSMs can successfully approximate the data distribution. We will flag this more clearly in the revised manuscript.
>
> ---
> ## [P4] Additional baselines
>
> Thank you for highlighting this concern. We direct you to Fig 13 (App I) where we have included a subset of uncertainty-based AL methods. These results feature EPIG (from the information family in Table 1) and other uncertainty-based metrics (marginal entropy, mean-std) **[C2]**.
>
> **Actions taken:** We have made the links to these additional results more prominent in the main paper.

---

### Official Review · Reviewer_CwS1 · 2024-07-13

**Soundness:** 3
**Presentation:** 2
**Contribution:** 3
**Rating:** 7
**Confidence:** 2

**Summary:**

This paper introduces a novel problem setting in Active Learning (AL) called Partially Observable Cost-Aware Active-Learning (POCA). POCA deals with situations where acquiring both features and labels comes with a cost and where datasets are partially observed, meaning some features might be missing for certain data points. The authors propose µPOCA, an instantiation of POCA that uses Generative Surrogate Models (GSMs), specifically Large Language Models (LLMs), to impute missing features and estimate the uncertainty reduction from acquiring both features and labels. This uncertainty reduction then guides the acquisition process, aiming to maximize the model's generalization capability within a given budget. The authors theoretically show that acquiring both labels and features leads to greater uncertainty reduction compared to acquiring only labels (the traditional AL approach) and provide empirical validation of µPOCA's effectiveness on various synthetic tabular datasets.

**Strengths:**

- Originality: The paper tackles a practical problem in AL that has been largely overlooked - the case of partially observed data with associated feature and label acquisition costs. Formalizing POCA problem setting is on its own a valuable contribution.
- Significance: The paper provides theoretical justification for µPOCA, demonstrating its potential to outperform traditional AL methods in specific scenarios. The empirical results support the effectiveness of the proposed approach.
- Addressing imputation vs. acquisition: The authors address the important question of whether imputation can replace data acquisition, showing that while LLM imputation is beneficial, it doesn't reach the performance of acquiring all features (Figure 7).

**Weaknesses:**

- Clarity and flow: The paper's clarity and flow could be improved. For example, the use of GSMs for both training and metric estimation is not immediately clear and could be better explained upfront. The notation is sometimes confusing, for example, the use of 'j' to represent both a single feature index and a set of feature indices, as in 'ci,j' and 'xj', and J+1 being used for the label can be easily missed. Additionally, some crucial concepts, like the need for a stochastic GSM (P1 in Appendix F), are introduced late in the paper, making it difficult for the reader to grasp their importance. Finally, the difference between Scenario 1 and 2 could be better explained as it's not entirely clear what the different implications of the two are and how the unnatural pattern of Scenario 1 impacts the performance of µPOCA.
- Potentially strong assumption of independence: The theoretical justification seems to rely on the assumption of independence between generalization capability (G) and unobserved features (Xj) given observed data (•).  Intuitively it seems like this assumption would not hold in many real-world scenarios. The authors should discuss the practicality of the assumption, the implications of violating this assumption, and potential mitigation strategies.

**Questions:**

- Have you considered evaluating µPOCA on real-world datasets with varying degrees and patterns of missingness? This would provide a more comprehensive understanding of its practical applicability and limitations.
- Are there specific situations where µPOCA significantly outperforms traditional AL, even when the GSM imputation accuracy is not perfect?

**Limitations:**

The authors briefly address the limitations of µPOCA.

---

> ### Author Rebuttal · Authors · 2024-08-07
>
> *Thank you for your feedback. We have addressed your individual points in this response and in the revised manuscript*
>
> ---
> ## [P1.1] GSM for metric estimation and training
> ---
> Thank you for your suggestion; we acknowledge that this important detail should be clear.
>
> **Update.** In response, before line 165 (after the new inclusion of [P1.3]), we added: *"For models with fixed-size input, like Random Forest, GSMs can be used to impute missing features (see Section [3.2] for details)."*
>
>
> ## [P1.2] Clarity on indexes
> ---
> **Index omissions** Bold variables are defined as a set of variables in line 94, which we agree can be easily overlooked. To clarify, we have now highlighted "**Bold variables**" for emphasis.
>
> Additionally, we corrected line 149 by bolding j, and we found no other omissions. The omission of $i$ is noted in line 96; to make this clearer, we state ".... is omitted when unnecessary, i.e., $x_{i,j}\equiv x_{j}$".
>
> **Index J+1** is an important detail. We thank the reviewer for spotting it. Consequently, for improved clarity, now line 99 states: "...indexed by $\boldsymbol{j}$, for instance, $i$, with index $j = J+1$ indicating the label acquisition".
>
> ## [P1.3] Stochastic GSMs
> ---
> As the reviewer correctly mentions, the stochasticity of the GSM is a key in uPOCA. To be more direct about this, we made the following updates:
>
> - **Update 1**, after the presenting the functionality of GSMs, we include in line 165 the following: *"**Why generative imputation can help Active Learning?** Note, that various (unobserved) features can be compatible with the observed features $x_{\boldsymbol{o}}$, leading to potentially different outcomes. In this case, a traditional AL metric would select this instance because $x_{\boldsymbol{o}}$ alone is insufficient to predict $Y$. However, generative imputation incorporates information about unobserved features into the predictive model, allowing it to select instances that remain uncertain even after this information is considered. Additionally, generative imputation can identify features that are not worth acquiring, as changes to these features do not impact $Y$.*
>
> - **Update 2**, We include the canonical example (see attached PDF) complementing this explanation in Appendix K.
>
> ## [P2] Independence assumption
> ---
> Thank you for highlighting this concern, which has helped us to be more explicit in illustrating why this is **not a strong assumption**.
>
> Lines L176-181 state that the assumption is valid for the supervised learning models we consider. Appendix B explains this further, and Appendix H provides empirical results confirming the validity of this assumption through Eq. (7). We made two modifications to clarify this:
>
> - **Update 1.** Now sentence L176 reads: *"Note that the independence assumption of Proposition 1 **is not a strong assumption and is valid** for the supervised models we consider."*. Also, in L181, we corrected the link to Appendix H, which was previously incorrect.
> - **Update 2.** Now sentence L180 reads: "... detailed explanation of this independence assumption's validity. In the same Appendix, we include a more informal but intuitive and practical explanation". The intuitive and practical explanation  is the following:
> ***Intuition.** Let the RV of observed and unobserved features be $X_1$ and $X_2$, respectively. Consider $\mathcal{G} = \Omega$, representing the EIG case, where the model parameter is related to generalization. Our downstream model is a Bayesian neural network (BNN) that samples $w_1$ half the time and $w_2$ the other half. This BNN is trained on dataset $\mathcal{D}$, using samples from $[X_1, X_2]$ as input. Given an instance with observation $x_1$, we use GSM to sample $x^\*_2$. The prediction $y$ is obtained by sampling $w^\*$ from the BNN and $x^\*_2$ from GSM, resulting in $y^\* = w^\*([x_1, x^\*_2])$. It is clear to notice that knowing $w^\*$ gives **no information** about the possible values of $x^\*_2$; this is because how we sample $w^\*$ depends on $\mathcal{D}$ and how the model is trained. This is the **independence assumption**. It is only possible to know something about $x^\*_2$ when we know $y^\*$, thanks to $y^\* = w^\*([x_1, x^\*_2])$. In the **observation**, we refer to this as the immorality.*
>
> ## [Q1] Study on different missingness patterns ##
> ---
>
> We believe that researching pattern missingness and its impact on GSM and acquisition performance is an important area of study that deserves a comprehensive study on its own. That said, we agree that a better understanding of how partial observability affects the acquisition process is necessary. In our study, we already consider real-world datasets that were engineered to construct partial observability. However, to be able to fully control the level of correlation of unobserved and observed feature, we extend this study using synthetic datasets in **[G2]**.
>
> ## [Q2] when $\mu$POCA outperform traditional AL under limited GSMs
> ---
>
> Active Learning's main limitation is that its acquisition metrics do not consider uncertainty reduction in feature acquisition. Our work addresses this by using GSMs to estimate the distribution of unobserved features and samples from this distribution to estimate the uncertainty reduction, though, as noted by reviewers, the GSM can be limited in some applications. uPOCA-based methods offer the advantage of approximating unobserved features, and even if imperfect, it can help identify non-useful features. To determine which unobserved features are relevant, we only need to observe some effect in the outcome when varying these features. In contrast, irrelevant features will have minimal impact on the outcome despite variability.

---

> > ### Comment · Reviewer_CwS1 · 2024-08-08
> >
> > Thank you for the detailed clarification, especially regarding the independence assumption. After considering the other reviews and your response, I believe the combined impact and the applicability of this work warrant a higher rating of 7.

---

> > > ### Author Response · Authors · 2024-08-09
> > > **Thank you**
> > >
> > > Thank you for your thoughtful feedback and for positively acknowledging our rebuttal. We appreciate your insights, which have been instrumental in improving the quality of our work.

---

### Official Review · Reviewer_ci71 · 2024-07-13

**Soundness:** 3
**Presentation:** 3
**Contribution:** 2
**Rating:** 6
**Confidence:** 4

**Summary:**

This paper  addresses the challenge of efficiently gathering features and labels in partially observed settings to enhance model generalization while considering data acquisition costs. It introduces POCA and its Bayesian instantiation, leveraging Generative Surrogate Models (GSMs) to impute missing features and compute uncertainty-based active learning metrics. The paper demonstrates the practical usage of µPOCA through empirical validation across different datasets.

**Strengths:**

1.The paper effectively formalizes the problem of active learning for partially observed data, presenting a practical approach that leverages GSMs for imputing missing features. This makes the method applicable to real-world scenarios where data acquisition is costly and incomplete.
2.Utilizing LLMs as GSMs to impute missing features is a novel approach. This leverages the generative capabilities of LLMs to enhance the estimation of uncertainty reduction metrics, which is crucial for the proposed method's success.

**Weaknesses:**

1.While the practical approach is well-formalized, the theoretical contributions and novelty are somewhat limited. The problem addressed by POCA has similarities with experimental design, and the extension of existing active learning methods to partially observed data is a relatively straightforward adaptation.
2.The paper does not adequately address how partial observability of features impacts the overall performance and reliability of the predictive model. There is a need for a deeper exploration of the theoretical implications and limitations of using GSMs for feature imputation.
3.In Equation 1, it does not seem to require that the data samples' labels must be annotated. If a sample only reveals a few features, how does such annotation help supervised learning? The only benefit I can think of is for the generative model, but the authors did not explain this clearly.
4.The description of the predictive model in Equation 3 is inaccurate. According to the authors' definition and the footnote on page 3, bold bold x represents fully observed features, but the predictive model should generally accept partially observed data.
5.The meaning of p_{phi} is unclear to me. In line 92, the authors mention that phi is a model we can employ. Does this mean that p_{phi}() represents a specific density function (e.g., Gaussian)? If so, how can we ensure that both the likelihood and posterior follow the same distribution, whose density is denoted by p_{phi}? The authors did not impose any conjugate restrictions.

**Questions:**

In Equation 1, it appears that the annotation strategy does not require that data samples' labels be annotated. If a sample only reveals a few features, how does such annotation benefit supervised learning? Is the primary benefit for the generative model? If so, could you provide a clearer explanation of this benefit?

---

> ### Author Rebuttal · Authors · 2024-08-07
>
> *We are grateful to the reviewer for their insightful feedback that has improved the paper.*
>
> ---
> ## [P1] Emphasizing novelty
> Thank you for highlighting this concern. Allow us to further clarify our novelty and theoretical contributions for partially observable data acquisition.
>
> **Theoretical analysis.** We start by focusing on $\mu$POCA, which we theoretically analyze in Proposition 1, applying to a family of Bayesian methods. Although the theorem may initially seem intuitive, it crucially implies that the new acquisition term requires estimating the *distribution* of the unobserved features. This requirement led us to develop *GSMs* to estimate this distribution. In other words, this theoretical rationale, which begins by analyzing uncertainty reduction in a partially observable setting, highlights the necessity of GSMs in this context, representing a novel contribution.
>
> **Methodological novelty.** To demonstrate the methodological novelty of $\mu$POCA, we compare it with **Vanilla Active Learning (VAL)** and **Active Learning with Imputation (ALI)** using EIG as an illustrative example:
> * **PO-EIG.** The acquisition metric is represented by $I(\Omega, Y|x_{\boldsymbol{o}}, \mathcal{D}, X_j)$, which measures predictive uncertainty considering the variability of $X_j$. This helps identify which values of $X_j$ are worth acquiring, as its *variability* allows measuring the impact of uncertainty reduction on predictive performance. `Fig 3a` in the attached PDF illustrates this, showing that exploring the unobserved regions $X_2$ and $X_3$ helps understand areas of uncertainty in the predictive model. This capability enables us to measure *what features are relevant* and in *what regions*.
> * **VAL**, **ALI**. We use $\tilde{x}$ to denote an imputation, the acquisition metrics for VAL and ALI are then represented by $I(\Omega, Y|x_{\boldsymbol{o}}, \mathcal{D})$ and $I(\Omega, Y|x_{\boldsymbol{o}}, \mathcal{D},\bar{x})$, respectively. *These metrics cannot determine which features to acquire*: VAL does not consider unobserved features and their impact on predictive performance; ALI provides only a point estimate in the unobserved region (as shown in `Fig 3a` from the PDF), which is insufficient for understanding uncertainty in the unobserved space.
>
> **Actions taken:** This discussion and the illustrative example in `Fig 3` have been added to App B.
>
> ---
> ## [P2] Effect of partial observability on performance
> We appreciate your feedback and agree that partial observability (PO) is a critical factor that can influence the ability of GSMs to accurately approximate the underlying data distribution. This approximation becomes vital as it directly impacts both predictive accuracy and acquisition performance in downstream tasks.
>
> **Actions taken:** We hypothesize that PO can affect acquisition performance when the GSM does not accurately learn the underlying data distribution, which is affected by (1) the approximation power of the GSM, and (2) the intrinsic properties of the partial observability. To investigate our hypothesis, we conducted a comprehensive investigation in **[G1, G2]** of the global response, highlighting that GSM approximation performance and intrinsic correlations in the dataset affect $\mu$POCA performance.
>
> ---
> ## [P3] Acquisition of only features in POCA
> Thank you for raising this point. As you correctly identified, Eq 1 describes that acquiring labels or features can improve generalization. We intended for this formalism to be as general as possible, as our work is the first to formalize the POCA problem.
>
> We clarify that in a purely *supervised learning* setting, acquiring only features does not improve generalization performance. However, this does not preclude this application for other learning problems. Two prominent examples are *semi-* and *self-supervised* learning, where models use additional unlabeled data to enhance generalization (by learning underlying structures).
>
> **Actions taken:** We have clarified our statement in L100 and explicitly referenced applications of Eq 1 in semi and self-supervised learning.
>
> ---
> ## [P4] Clarifying Eq 3
> Thank you for spotting this. We agree with the reviewer that in Eq 3, the input of the predictive model can be partially observed data.
>
> **Actions taken.** We have updated Eq 3 to describe the predictive model as $p_{\phi}(y|\mathbf{x}')$, where $\mathbf{x}'$ represents partially observed data. We further clarify that while models that accept variable-length inputs (eg Transformer, random forests) naturally handle this, models that expect fixed-size inputs would require imputation.
>
> ---
> ## [P5] Clarifying $\phi$
> Effectively, $p_{\phi}(y|\boldsymbol{x})$ represents a likelihood over $Y$ when conditioned on the input $\boldsymbol{x}$. $\phi$ represents the model, which is specified by the functional form, the parameters $\omega$, and its distributions. For example, in a Bayesian neural network:
>
> $$p_{\phi}(y|\mathbf{x})=\int_{\Omega}p(\omega|\mathcal{D})p_{\phi}(y|\mathbf{x}, \omega) \, d\omega$$
>
> Where $p(\omega|\mathcal{D})$ is the posterior distribution over weights, $p(\omega|\mathcal{D}) \propto p(\mathcal{D}|\omega)p(\omega)$. This model is specified with an architecture and priors over weights, $p(\omega)$. As such, $p_\phi(y|\mathbf{x})$ is the posterior predictive distribution marginalized over $\omega$.
>
> We note that the posterior $p(\omega|\mathcal{D})$ does not necessarily have the same distribution as the likelihood $p_\phi(y|\mathbf{x})$ (eg in BNNs, $p(\omega|\mathcal{D})$ are typically Gaussian, but the likelihood function could be categorical for classification). As the reviewer has pointed out, obtaining an analytical expression might require the use of conjugate distributions. However, in many practical cases, this is not necessary (eg by using variational inference).
>
> **Actions taken:** We have clarified the meaning of $\phi$ in Eq 3 using a paraphrased version of this discussion.

---

> > ### Comment · Reviewer_ci71 · 2024-08-08
> > **Rebuttal response - POCA**
> >
> > Thank you for the detailed rebuttal. Most of my concerns are addressed. I want to adjust my rating to weak accept.

---

> > > ### Author Response · Authors · 2024-08-09
> > > **Thank you**
> > >
> > > Thank you for your valuable feedback. We are glad to have addressed your concerns and appreciate your insights, which have significantly enhanced the quality of our work.

---

### Official Review · Reviewer_tNwd · 2024-07-13

**Soundness:** 3
**Presentation:** 3
**Contribution:** 2
**Rating:** 6
**Confidence:** 4

**Summary:**

This paper introduces a novel active learning framework for optimizing data acquisition in scenarios with partially observed features and high costs. The proposed method, POCA utilizes GSM, specifically LLMs, to impute missing features and improve active learning metrics. By integrating Bayesian methods to maximize uncertainty reduction, the proposed active learning method can select features and labels to enhance predictive performance while minimizing acquisition costs.

**Strengths:**

The task itself is novel, optimizing the acquisition of data (features and labels) in scenarios where information is partially observed and the cost of acquiring additional data is high. The author defines the cost of active learning from a new aspect, it is interesting.

Additionally, the combination of GSMs (especially LLMs) to impute missing features is also a novel aspect.

**Weaknesses:**

1. The Monte-Carlo-based approach in POCA would be slow and with high computational cost due to its reliance on repeated random sampling to estimate quantities of interest.
2. This model is limited to tabular data.
3. Sometimes the performance of POCA is not as good as random sampling, POCA may vary depending on the specific characteristics of the dataset and the underlying correlations​.
4. Please consider more active learning methods as baselines except for BALD and random sampling.

**Questions:**

There are many typos, the author should carefully check the full manuscript. For example, Line 721 "?." and line 723 "Figure ??".

**Limitations:**

The authors have adequately addressed the limitations and potential negative societal impact of their work.

---

> ### Author Rebuttal · Authors · 2024-08-07
>
> *We are grateful to the reviewer for their insightful feedback that has improved the paper.*
>
> ---
> ## [P1] Computational costs of MC sampling
>
> Thank you for highlighting this concern. We agree that it is important to discuss $\mu$POCA's computational costs. We approach this discussion by **(1)** providing a detailed algorithm for complete acquisition in `Algorithm 1` (in the attached PDF), **(2)** detailing the complexity below, and **(3)** discussing how this cost can be alleviated in future works.
>
> **Training process.** Before we discuss inference complexity, we note that GSMs will be trained using available unlabeled data, which may be either fully observed (if a bank of fully observed unlabeled data is available) or partially observed (using the pool set itself). After the GSM is trained, it is used for inference during the acquisition process.
>
> **Inference complexity.** Using notation consistent with the paper, $I$ is the number of instances, $J$ is the number of features, and $S$ is the number of Monte-Carlo samples. A direct acquisition would consider selecting subsets of features over all instances, resulting in a complexity of $\mathcal{O}(I\*J^2\*S)$. To reduce this overhead, we introduce an approximate algorithm (detailed in S4.3), where we **(a)** first select the most informative instance (assuming all features are acquired), then **(b)** select the subset of features of that instance to acquire:
> * **(a)** As we now make more explicit in `Alg 1` (in attached PDF), the first step involves identifying the instance to acquire, which has a sampling cost of $\mathcal{O}(I\*S)$ (note, all unobserved features are sampled jointly).
> * **(b)** Once the instance is selected, the cost of acquiring a subset of features is bounded by $\mathcal{O}(J^2\*S)$.
>
> Our approximate procedure thus has a total complexity of $\mathcal{O}(I\*S + J^2 \*S)$, compared to  $\mathcal{O}(I\*J^2\*S)$ of the exact procedure. This analysis indicates that the dominant factor is the number of features $J$ as it affects sampling costs quadratically.
>
> **Future work.** In our work, we estimate uncertainty using Monte Carlo samples. However, a promising area for further research is improving efficiency by avoiding sampling with tractable analytical expressions. For example, assuming that features and labels are jointly Gaussian allows us to compute Eq. 3 in closed form. Other conjugate distributions can also be explored. Another approach to enhancing the efficiency of sampling-based approaches is in-context learning, as generative imputation can be done in parallel for multiple instances.
>
> **Actions taken:** The above analysis and `Alg 1` (attached PDF) have been included in App J, with appropriate references in the main text.
>
> ---
> ## [P2] Focus on tabular data
>
> We agree that the primary applications of POCA are in tabular data, and we focused our empirical analysis on this domain. However, we emphasize that tabular data is ubiquitous in the real world, particularly in fields such as finance, healthcare, industrial engineering, and retail, *where the POCA problem is most likely to be applicable.*
>
> However, our formalism and method can be readily extended to other modalities, such as images, by treating pixels or pixel patches as features. Although this is less common, practical applications could exist in fields like medical and satellite imaging, where occlusion caused by noise is prevalent. In these cases, it's crucial to determine when a sample requires additional information (features) for accurate prediction and model training. Similarly, interactive robots that learn through vision can also benefit from this approach.
>
> **Actions taken:** We have **(1)** added discussions on extensions to other modalities in the Future Works section (S5) and **(2)** extended Table 1 to include example applications involving other modalities, demonstrating the versatility of POCA.
>
> ---
> ## [P3] Conditions that affect POCA performance
>
> Thank you for highlighting this concern. We acknowledge that the performance of $\mu$POCA-based methods can vary in certain scenarios. The key question we want to answer is: in what settings do $\mu$POCA methods perform poorly? We hypothesize that this can occur when the GSM does not accurately learn the underlying data distribution, which is affected by (1) the approximation power of the GSM, and (2) the intrinsic properties of the partial observability.
>
> To investigate this question and our hypothesis, we conducted a comprehensive investigation in **[G1, G2]** of the global response, highlighting how GSM approximation performance and intrinsic correlations in the dataset affect $\mu$POCA performance.
>
> **Actions taken:** Thank you for this suggestion, which has improved our analysis of $\mu$POCA performance in different settings. We have introduced a dedicated section in the appendix for **[G1, G2]**.
>
> ---
> ## [P4] Additional baselines
>
> Thank you for highlighting this concern. We direct you to Fig 13 (App I) where we have included a subset of uncertainty-based AL methods. These results feature EPIG (from the information family in Table 1) and other uncertainty-based metrics (marginal entropy, mean-std) **[C2]**.
>
> **Actions taken:** We have made the links to these additional results more prominent in the main paper.
>
> ---
>
> *We hope that most of the reviewer’s concerns have been addressed. We’d be happy to engage in further discussions.*

---

> > ### Comment · Reviewer_tNwd · 2024-08-11
> > **reponse**
> >
> > I'm satisfied with your response and will raise my score to 6.

---

> > > ### Author Response · Authors · 2024-08-14
> > > **Thank you**
> > >
> > > Thank you for your insightful feedback. We’re pleased that we could resolve your concerns and are grateful for your input, which has greatly improved the quality of our work.

---

### Author Rebuttal · Authors · 2024-08-07

---

*We are grateful to the reviewers for their insightful feedback.*

We are happy to hear that reviewers found our work as a "novel task, optimizing the acquisition of data (features and labels) in scenarios where information is partially observed and the cost of acquiring additional data" (**tNwd**), which is noted as "a practical problem in AL that has been largely overlooked" (**CwS1**). Furthermore, we are also grateful for the acknowledgment of the formalization of POCA (**ci71**), noted as "on its own a valuable contribution" (**CwS1**).

We appreciate the recognition of the innovative combination of GSMs (particularly LLMs) to impute missing features, which was highlighted as a novel aspect (**ci71, tNwd**), and provides "theoretical justification for $\mu$POCA, demonstrating its potential to outperform traditional AL methods" (**CwS1**). The reviewers also highlighted that our work "addresses an interesting problem and proposes a simple and intuitive methodology" and is supported by "empirical analysis" (**A8wS**).

We have also taken the reviewers’ feedback into account and made the following key changes to improve the paper:
* **[G1] GSM impact on acquisition performance:** Demonstrating how underlying GSM performance affects acquisition results (see `Fig 1` in attached PDF), observing that GSM quality has a notable impact on acquisition performance.
* **[G2] Performance across varying data characteristics.** Analyzing $\mu$POCA's performance using synthetic datasets with varying correlations (see `Fig 2`), confirming our hypothesis that in settings with increased correlations between observed and unobserved features, the performance gains of POCA are more notable.

We hope these updates address the reviewers' concerns. We welcome any further feedback.

With thanks,

The Authors of #15686

---
## Overview
The performance of $\mu$POCA in partially observed settings fundamentally depends on how well the GSM approximates the distribution of the unobserved features. There are two key factors that affect the quality of this estimation:

1. **GSM's approximation power:** referring to the model's capacity to accurately model the unobserved features.
2. **Intrinsic characteristics of the dataset:** referring to inherent correlations between observed and unobserved features. Indeed, lower correlations is more challenging.

In what follows, we investigate each factor in turn, **[G1]** studying the impact of different GSMs and **[G2]** investigating different dataset characteristics. This approach aims to delineate the conditions under which $\mu$POCA is expected to excel.

---
## [G1] GSM impact on acquisition performance
**Setup.** We evaluated various LLMs (including `Mistral-7B-Instruct-v0.3`, `Phi-3`, and `Llama-3`) as GSMs to assess how model quality affects acquisition performance.

**Analysis.** `Fig 1` demonstrates that GSM quality significantly influences acquisition results. Notably, Mistral-7B consistently outperforms the alternative GSMs, with one exception in the housing dataset. Interestingly, Llama-3 performs well on this benchmark, highlighting this inter-model variability. The Cardio dataset further highlights these differences, with Mistral-7B performing significantly better than the other models.

**Takeaway.** These findings underscore the critical role of GSM quality in acquisition performance.

---
## [G2] Performance across varying data characteristics

**Setup.** Next, we turn our attention to investigating how the the data distribution affect acquisition performance. We are particularly interested in analyzing the effect of the correlation between unobserved features $X_{unobs}$ and observed features $X_{obs}$ on acquisition performance.

To demonstrate this, we examine a scenario where (1) $X_{unobs}$ correlate with the outcome, while (2) observed features do not. In this context, the GSM becomes crucial for downstream performance, as $X_{obs}$ alone provide insufficient information to predict outcomes accurately. As such, the GSM must effectively model the relationship between $X_{obs}$ and $X_{unobs}$ to acquire missing features critical for predicting the outcome.

We model both  $X_{obs}$ and $X_{unobs}$ as two-dimensional random Gaussian variables centered at zero and establish a specific **controllable** correlation between them through $\rho$:

$$\Sigma = [I\_2  ~~~~~~ \rho\_{X\_{obs}X\_{unobs}} I\_2 $$

$$~~~~~~ \rho\_{X\_{obs}X\_{unobs}}I\_2~~~~~~~~  I\_2 ] $$

The label $Y$ is then constructed to be independent of $X_{obs}$ using the orthogonalization:

$$X_{\text{orthogonal}} = X_{unobs} - X_{obs} (X_{obs}^T X_{obs})^{-1} X_{obs}^T X_{unobs}$$

which we use to construct the label using $\text{logits} = \frac{1}{1 + e^{-\sum B_{\text{orthogonal}}}}, \quad C = \mathbf{1}_{\text{logits} > \text{0}}$

**Analysis.** `Fig 2a` illustrates how varying $\rho$ between  $X_{obs}$ and $X_{unobs}$ *empirically* affects variable correlation, c, validating our synthetic experiment design. `Fig 2b` analyzes EIG (traditional active learning without GSM) and PO-EIG with varying $\rho$. We note that when the correlation between $X_{obs}$ and $X_{unobs}$ is low, GSMs provide no performance benefits. However, as correlation increases, the performance gains of PO-EIG over EIG expand significantly, confirming our hypothesis.

---
### References

**[C1]** Borisov, V., Leemann, T., Seßler, K., Haug, J., Pawelczyk, M., & Kasneci, G. (2021). Deep Neural Networks and Tabular Data: A Survey. IEEE Transactions on Neural Networks and Learning Systems, 35, 7499-7519.

**[C2]** Tharwat, A., & Schenck, W. (2023). A survey on active learning: State-of-the-art, practical challenges, and research directions. Mathematics, 11(4), Article 820.

**[C3]** Tom Rainforth. Adam Foster. Desi R. Ivanova. Freddie Bickford Smith. "Modern Bayesian Experimental Design." Statist. Sci. 39 (1) 100 - 114, February 2024. <!--https://doi.org/10.1214/23-STS915 -->

---

### Decision · Program_Chairs · 2024-09-25

**Decision:**

Accept (poster)

**Comment:**

The paper introduces 𝜇POCA, a novel framework for optimizing data acquisition in scenarios with partial observability and high acquisition costs. 𝜇POCA, the proposed instantiation, leverages Generative Surrogate Models (GSMs), particularly Large Language Models (LLMs), to impute missing features and maximize uncertainty reduction during acquisition. The reviewers agree that the problem addressed is timely and that integrating GSMs for feature imputation is a novel contribution with practical relevance.

However, the reviewers also highlight several areas for improvement. The reliance on Monte-Carlo sampling introduces significant computational costs, and the paper's clarity could be enhanced. Additionally, the empirical validation is primarily focused on tabular data (in the original submission), limiting the generalizability of the findings. The authors have provided a rebuttal addressing most of these concerns. If accepted, the authors are encouraged to refine the theoretical analysis and broaden the scope of their experiments.